# UPF1 plays critical roles in early B cell development

Noriki Iwai [1,2], Kotaro Akaki [1,2], Fabian Hia[1], Wei Li [1], Masanori Yoshinaga [1], Takashi Mino [1] & Osamu Takeuchi [1]

The ATP-dependent RNA helicase UPF1 plays a crucial role in various mRNA degradation pathways, most importantly in nonsense-mediated mRNA decay (NMD). Here, we show that UPF1 is upregulated during the early stages of B cell development and is important for early B cell development in the bone marrow. B-cell-specific *Upf1* deletion in mice severely impedes the early to late LPre-B cell transition, in which $V_H$-$D_H J_H$ recombination occurs at the *Igh* gene. Furthermore, UPF1 is indispensable for $V_H$-$D_H J_H$ recombination, without affecting $D_H$-$J_H$ recombination. Intriguingly, the genetic pre-arrangement of the *Igh* gene rescues the differentiation defect in early LPre-B cells under *Upf1* deficient conditions. However, differentiation is blocked again following *Ig light chain* recombination, leading to a failure in development into immature B cells. Notably, UPF1 interacts with and regulates the expression of genes involved in immune responses, cell cycle control, NMD, and the unfolded protein response in B cells. Collectively, our findings underscore the critical roles of UPF1 during the early LPre-B cell stage and beyond, thus orchestrating B cell development.

B lymphocyte development takes place in the bone marrow (BM) through the sequential events involving proliferation and DNA recombination at the V(D)J region of the *immunoglobulin* (*Ig*) *heavy chain* (*Igh*) locus and the VJ region of the *Ig light chain* locus. This process results in the generation of a diverse array of peripheral antibodies, culminating in the establishment of humoral immunity[1–3]. The process begins with the D-J recombination at the *Igh* locus ($D_H$-$J_H$) during the pro-B stage, followed by the $V_H$-$D_H J_H$ recombination occurring at the early large pre (eLPre)-B stage[4]. In conjunction with surrogate light chains, the successfully recombined *Igh* gene facilitates the expression of pre-B cell receptor (BCR) at the late large pre (lLPre)-B stage. This event promotes cell survival and proliferation, as well as V-J recombination at the *Ig light chain* ($V_L$-$J_L$) locus during the small pre (sPre)-B stage. Upon the successful recombination at the *Ig light chain* locus, B cells express their own BCR to progress to the immature (Imm)-B stage. Throughout B lymphopoiesis in the BM, these cell state transitions are orchestrated by sequential gene expression programs and harmonious control of transcriptional factors, including E2A, EBF1, and Pax5[5].

In addition to transcriptional control of gene expression, post-transcriptional mRNA decay programs emerge as indispensable players in sculpting the transcriptomic landscape, a critical role during B cell development[6]. A dynamic interplay of RNA-binding proteins (RBPs) is initiated by their recognition of *cis*-elements within target mRNAs. This recognition paves the way for the recruitment of ribonucleases that execute mRNA degradation through processes such as deadenylation or endoribonuclease activities. Among the array of RBPs, the Zinc finger protein 36 (ZFP36) family members, ZFP36L1 and ZFP36L2, emerge as key players in promoting V(D)J recombination. They achieve this by preserving cell quiescence before pre-BCR expression, primarily through their recognition of AU-rich element-containing mRNAs[7]. The splicing regulators TIA1 and TIAL1 play a pivotal role in B cell development by inducing the expression of the DNA damage repair machinery[8]. In addition, post-transcriptional mRNA $N^6$-methyladenosine (m6A) modification catalyzed by is required for the large-pre-B-to-small-pre-B transition, emphasizing the multi-tiered significance of post-transcriptional regulation during the

[1]Department of Medical Chemistry, Graduate School of Medicine, Kyoto University, Kyoto, Japan. [2]These authors contributed equally: Noriki Iwai, Kotaro Akaki. ✉e-mail: tmino@mfour.med.kyoto-u.ac.jp; otake@mfour.med.kyoto-u.ac.jp

B cell development[4]. The components of RNA decay machinery, such as CNOT3 within the carbon catabolite repression 4-negative on TATA-less (CCR4-NOT) deadenylation complex, are essential for V(D)J recombination via the suppression of p53 expression[9,10]. Furthermore, elements of RNA exosome, another ribonuclease complex, contribute by regulating proper B lymphopoiesis during the pro-B cell to large pre-B cell transition, primarily through the degradation of long non-coding RNAs[11,12]. With the accumulating knowledge in this field, B cell differentiation serves as an ideal system for the in vivo assessment of the roles played by post-transcriptional regulation.

Upstream frameshift 1 (UPF1) is an ATP-dependent RNA helicase involved in various mRNA degradation pathways, including nonsense-mediated mRNA decay (NMD), Staufen-mediated mRNA decay (SMD), and Regnase-1-mediated mRNA decay. Mechanistic roles of UPF1 were most extensively characterized in NMD, which is triggered by translation termination at a premature termination codon (PTC), leading to the formation of SMG1-UPF1-eRF1-eRF3 (SURF) complex[13–15]. Subsequently, the kinase SMG1 phosphorylates UPF1 upon interaction with the exon-junction complex (EJC), and the phosphorylated UPF1 (p-UPF1) recruits the endoribonuclease SMG6 and adapter proteins SMG5 and SMG7, which in turn recruit XRN1 or the CCR4-NOT complex for target mRNA degradation[14,16–18]. UPF1 is also involved in the degradation of immune-related mRNAs by RNase Regnase-1, which forms a complex with UPF1 to degrade target mRNAs[19–21]. Thus, UPF1 is not only involved in surveilling aberrant mRNAs but also regulates the decay of a set of cellular mRNAs.

While the in vitro mechanisms of UPF1-mediated mRNA metabolism are well-characterized, its in vivo role has remained less known. In this study, we generate B cell-specific Upf1 knockout mice, allowing us to explore the cell-intrinsic function of UPF1 in B cell development. Thus, we follow this process from early to late LPre-B cells and through the sPre- to the immature B stage. UPF1 interacts with and controls the expression of genes involved in immune responses, cell cycle regulation, and unfolded protein response (UPR) in B cells. Collectively, this study provides mechanistic insight into the complex in vivo role of UPF1 in B cell differentiation.

# Results

## UPF1 expression increases in early LPre-B cells and late LPre-B cells

We first examined the expression level of UPF1 at distinct stages of B cell development using mouse BM cells (Fig. 1a and Supplementary Fig. 1a). Interestingly, the UPF1 protein expression was remarkably upregulated in early LPre- and late LPre-B cells compared to pre-pro-, pro-, sPre-, Imm-, and recirculating (Rec)-B cells (Fig. 1b). Furthermore, phosphorylation of UPF1 at S1111 residue, which is known to be mediated by SMG1 during initiation of NMD[22], was highly increased in early LPre and late LPre-B cells. Conversely, the mRNA expression levels of Upf1 remained relatively consistent across the various stages of B cell differentiation (Fig. 1c), suggesting that the UPF1 protein is controlled translationally and/or post-translationally during B cell development. These data indicate that both the expression and activity of UPF1 dynamically change during B cell development.

## UPF1 is essential for guiding B cell differentiation from early to late LPre-B stage

To investigate the role of UPF1 during B cell development, we generated Upf1 flox mice and crossed them with Mb1-Cre mice to achieve B cell-specific Upf1 depletion (Supplementary Fig. 1b–d). In Upf1 flox mice, two loxP sites are flanking exons 4-6 of the Upf1 gene, resulting in the suppression of the UPF1 helicase domain expression upon the expression of Cre recombinase (Supplementary Fig. 1b and c). The size of the spleen and the number of splenocytes in $Upf1^{Flox/Flox}$ $Mb1^{Cre/+}$ (Upf1-cKO) mice were noticeably smaller than that of control (Ctrl) mice (Fig. 1d and e). B cells were rarely found in the spleen of Upf1-cKO

mice (Fig. 1f and g), indicating that UPF1 is essential for maintaining a mature B cell population in secondary lymphoid organs.

We next examined the effect of UPF1 depletion on B cell differentiation in the BM. We confirmed that UPF1 protein expression was abolished at Pro-B and early LPre-B stages in Upf1-cKO mice (Fig. 1h). Flow cytometry analysis revealed that late LPre-, sPre-, Imm-, and Rec-B cell populations vanished in Upf1-cKO mice, while the numbers of pro-B and early LPre-B cells remained comparable between Ctrl and Upf1-cKO mice (Fig. 1i and j). These findings offer compelling evidence that UPF1 depletion obstructs the differentiation process from early to late LPre-B cells, a pattern that aligns with the observed upregulation of UPF1 expression in this developmental stage (Fig. 1b). The primary distinction between early LPre- and late LPre-B cells is that early LPre-B cells undergo $V_H$-$D_H J_H$ recombination, whereas late LPre-B cells bear the recombined Igh gene (Fig. 1a). To examine whether the interruption of B cell differentiation in Upf1-cKO mice occurs during the $V_H$-$D_H J_H$ recombination phase, we stained the BM B cells with cell surface markers, c-kit and CD25, which are highly expressed before and after $V_H$-$D_H J_H$ recombination, respectively. We rarely detected CD25+ B cells in the Upf1-cKO BM (Fig. 1k), ascertaining that UPF1 is essential to establish Igh-recombined cells during B cell differentiation. These results suggest that UPF1 is required for the transition from early to late LPre-B cell stage, characterized by $V_H$-$D_H J_H$ recombination.

## Transcriptome-scale profiling of early LPre-B cells lacking Upf1

Considering the essential role of UPF1 in the differentiation from early to late LPre-B stage, we examined the gene expression profile of early LPre-B cells from Upf1-cKO and Ctrl mice through RNA sequencing (RNA-seq) analysis (Fig. 2a and Supplementary Data 1). Our analysis revealed that 540 genes exhibited significantly higher expression levels (Log$_2$ fold change (FC) > 2, adjusted P (adj. P) < 0.05), and 201 genes displayed reduced expression (Log$_2$FC < −2, adj. P < 0.05) in Upf1-deficient early LPre-B cells (Fig. 2a and Supplementary Data 1). Prominently recognized NMD target genes, including Gas5, Gadd45b, and Ddit3, exhibited elevated expression in Upf1-deficient cells, suggesting a compromised NMD mechanism within Upf1-deficient early LPre-B cells (Fig. 2a). In addition, gene set enrichment analysis (GSEA) using hallmark gene sets revealed enrichment of gene sets associated with interferon (IFN) responses among highly expressed genes in Upf1-deficient cells (Fig. 2b). Notably, IFN-stimulated genes (ISGs), such as Mx2, Ifi44, and Oasl1, were highly expressed in Upf1-deficient early LPre-B cells (Fig. 2c, d). Conversely, genes linked to E2F TARGETS, OXIDATIVE PHOSPHORYLATION, MYC TARGETS V1, and G2M CHECKPOINT showed decreased expression in Upf1-deficient cells (Fig. 2e, f and Supplementary Fig. 2a–f). Moreover, GSEA using all gene sets revealed that Upf1-deficient cells expressed diminished levels of the genes associated with adaptive immune responses and cell cycle, including Igh-derived transcripts and histone mRNAs, respectively (Fig. 2g–i and Supplementary Fig. 2g, h). These findings collectively underscore the protective role of UPF1 in preventing aberrant expression of IFN signature genes while emphasizing the dependency on UPF1 for the expression of Igh-derived transcripts and genes associated with the cell cycle.

## Lack of IFN receptor failed to rescue the B cell development in Upf1-cKO mice

Since type I IFNs are reported to impede pro-B cell development[23], we hypothesized that dysregulated expression of ISGs might disrupt the proper differentiation of B cells in Upf1-cKO mice. Therefore, we sought to investigate whether the inhibition of IFN signaling could ameliorate the defects in B cell development in Upf1-cKO mice by crossing IFN receptor (Ifnar1)-KO mice with Upf1-cKO mice. However, the spleen size in Upf1/Ifnar1 double knockout (DKO) mice ($Upf1^{Flox/Flox}Mb1^{Cre/+}Ifnar1^{-/-}$) remained diminutive, akin to that of Upf1-cKO mice (Fig. 3a). Furthermore, splenic B cell populations were severely

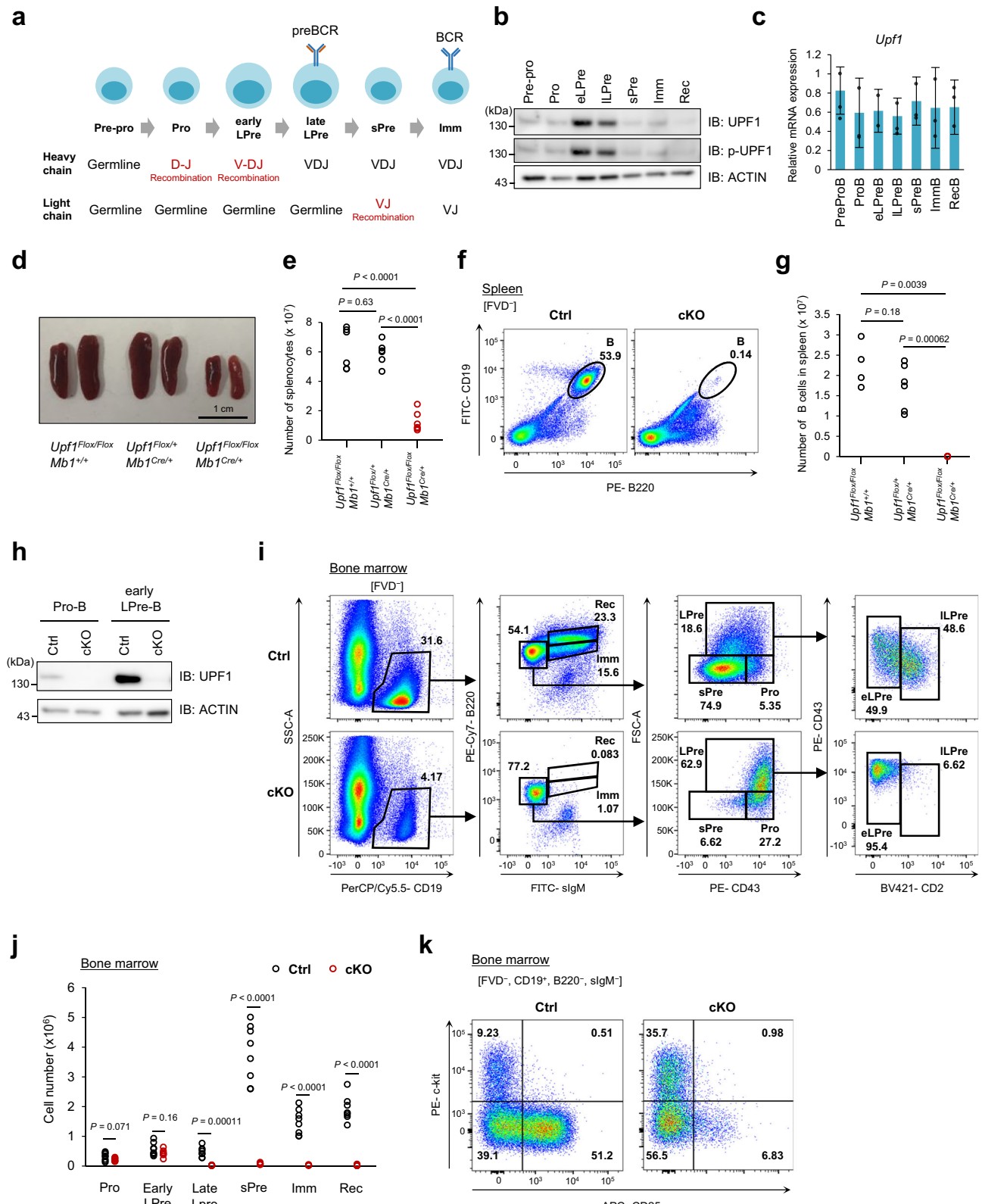

diminished in DKO mice, comparable to the observations in *Upf1*-cKO mice (Fig. 3b). Notably, *Upf1/Ifnar1* DKO mice showed B cell differentiation defects at early to late LPre-B transition, mirroring the patterns observed in *Upf1*-cKO mice (Fig. 3c and d). The analysis of early LPre-B cells in DKO mice revealed a reduction in ISG expression, confirming the suppression of IFN signaling (Fig. 3e). Noteworthy, we observed no apparent difference in the B cell development in the BM

between control (*Upf1^{Flox/Flox} Mb1^{+/+} Ifnar1^{+/-}*) and *Ifnar1*-KO (*Upf1^{Flox/Flox} Mb1^{+/+} Ifnar1^{-/-}*) mice, suggesting that the loss of type I IFN signaling alone does not affect B cell development in the BM (Supplementary Fig. 3). Collectively, these data demonstrate that the differentiation defect from early LPre-B cells to late LPre-B cells observed in *Upf1*-cKO mice cannot be solely attributed to the heightened IFN signaling levels.

**Fig. 1 | UPF1 is essential for B cell differentiation from early LPre-B to late LPre-B cell. a** Schematic illustration of B cell development in mammalian BM. **b** Immunoblot showing protein expression and phosphorylation (S1111) of UPF1 at each stage of B cell differentiation in mouse BM. The B cell progenitors were identified as Pre-pro-B (Lineage$^-$, CD19$^-$, B220$^+$, sIgM$^-$), Pro-B (Lineage$^-$, CD19$^+$, B220$^+$, sIgM$^-$, CD43$^{hi}$, FSC$^{lo}$, CD25$^-$, CD2$^-$), early large pre (LPre)-B (Lineage$^-$, CD19$^+$, B220$^+$, sIgM$^-$, CD43$^+$, FSC$^{hi}$, CD2$^-$), late LPre-B (Lineage$^-$, CD19$^+$, B220$^+$, sIgM$^-$, CD43$^+$, FSC$^{hi}$, CD2$^+$), small Pre (sPre)-B (Lineage$^-$, CD19$^+$, B220$^+$, sIgM$^-$, CD43$^{lo}$, FSC$^{lo}$, CD25$^+$, CD2$^+$), immature (Imm)-B (Lineage$^-$, CD19$^+$, B220$^{lo}$, sIgM$^+$), and recirculating (Rec)-B (Lineage$^-$, CD19$^+$, B220$^{hi}$, sIgM$^+$). **c** mRNA expression of *Upf1* measured by RT-qPCR at each stage of B cell differentiation in mouse BM ($n = 3$, Each bar represents the mean ± SD from biological replicates). **d** Appearance of mouse spleens from indicated genotypes. **e** Numbers of splenocytes derived from indicated mice ($N = 6$ for each genotype). $P = 0.63$ (*Upf1*$^{Flox/Flox}$/*Mb1*$^{+/+}$ vs. *Upf1*$^{Flox/+}$/*Mb1*$^{Cre/+}$), $P = 0.000086$ (*Upf1*$^{Flox/Flox}$/*Mb1*$^{+/+}$ vs. *Upf1*$^{Flox/Flox}$/*Mb1*$^{Cre/+}$), $P = 0.00000082$ (*Upf1*$^{Flox/+}$/*Mb1*$^{Cre/+}$ vs. *Upf1*$^{Flox/Flox}$/*Mb1*$^{Cre/+}$) (**f**) Flow cytometry plots of B cells (B220$^+$, CD19$^+$) in the spleen of indicated mice. **g** The cell numbers of splenic B cells (B220$^+$, CD19$^+$) derived from indicated mice (*Upf1*$^{Flox/Flox}$ *Mb1*$^{+/+}$; $N = 4$, *Upf1*$^{Flox/+}$ *Mb1*$^{Cre/+}$; $N = 6$, *Upf1*$^{Flox/Flox}$ *Mb1*$^{Cre/+}$; $N = 6$). **h** The expression level of UPF1 proteins in pro- or early LPre-B cells derived from Ctrl or cKO mice. **i** Flow cytometry plots of indicated populations in the BM of indicated mice. FVD: Fixable Viability Dye. **j** Cell number of indicated populations in the BM of indicated mice. BM were harvested from bones of both hind legs ($N = 8$ for each genotype). $P = 0.071$ (Pro-B), $P = 0.155$ (eLPre-B), $P = 0.00011$ (lLPre-B), $P = 0.000012$ (sPre-B), $P = 0.000024$ (Imm-B), $P = 0.000037$ (Rec-B) (**k**) Flow cytometry plots of indicated cells derived from BM of indicated mice. Results are representative of at least three independent experiments. The *p*-values were calculated using a two-sided Student's *t* test. Source data are provided as a Source Data file.

## UPF1 is necessary for *Igh* V$_H$-D$_H$J$_H$ recombination during early LPre-B stage

The GSEA revealed a marked reduction in *Igh* transcripts in *Upf1*-deficient early LPre-B cells compared with Ctrl cells (Fig. 2g, h). Therefore, we further checked the expression of individual transcripts within the *Igh* locus. The majority of *Ighv* transcripts, which constitute the variable region, were found to be significantly diminished in *Upf1*-deficient early LPre-B cells (Fig. 4a). In contrast, transcript levels of *Ighm* (representing a constant region), *Ighj* (joining region), and *Ighd* (diversity region) remained unaltered following UPF1 depletion (Fig. 4b–d).

It is known that the expression of *Ighv* transcripts is induced during or after the V$_H$-D$_H$J$_H$ recombination[24]. Therefore, we next investigated whether UPF1 depletion has any impact on V$_H$-D$_H$J$_H$ recombination by performing genomic DNA PCR using three different primer sets: one for the detection of distal (V$_H$J558 to D$_H$J$_H$), another for proximal (V$_H$7183 to D$_H$J$_H$), and the third for D$_H$ to J$_H$ rearrangement (Fig. 4e). In Ctrl early LPre-B cells, all PCR products corresponding to D$_H$-J$_H$, distal V$_H$-D$_H$J$_H$, and proximal V$_H$-D$_H$J$_H$ rearranged genomic DNA were detected, indicating the occurrence of D$_H$-J$_H$ and V$_H$-D$_H$J$_H$ recombination (Fig. 4f). In contrast, *Upf1*-deficient early LPre-B cells exhibited an absence of both proximal and distal V$_H$-D$_H$J$_H$ rearranged bands, while D$_H$-J$_H$ rearrangement was still detectable under *Upf1* deficiency (Fig. 4f). These findings indicate that *Upf1* deficiency selectively inhibits V$_H$-D$_H$J$_H$ while leaving D$_H$-J$_H$ recombination unaffected. Accordingly, the protein expression of the intracellular Igμ heavy chain was rarely detected in CD43$^+$, CD2$^-$ B cells from *Upf1*-cKO mice (Fig. 4g). Collectively, these data demonstrate the requirement of UPF1 in facilitating V$_H$-D$_H$J$_H$ recombination at the early LPre-B stage, thereby enabling the proper expression of the Igμ heavy chain.

The aforementioned results motivated us to investigate whether introducing genetically pre-arranged *Igh* could rescue the impaired B cell development in *Upf1*-cKO mice. To address this, we crossbred *Upf1*-cKO mice with B1-8$^{hi}$ *Igh* pre-arranged mice[25] (hereafter referred to as cKO/*Igh*$^{B1-8hi}$ mice). The number of late LPre- and sPre- B cells was significantly increased in cKO/*Igh*$^{B1-8hi}$ mice compared with *Upf1*-cKO mice with WT *Igh* gene (cKO/*Igh*$^{WT}$) mice, with the number of late LPre-B cells notably similar between cKO/*Igh*$^{B1-8hi}$ and Ctrl/*Igh*$^{B1-8hi}$ mice (Fig. 4h, I and Supplementary Fig. 4a). Furthermore, c-Kit$^-$, CD25$^+$ B cells in cKO/*Igh*$^{B1-8hi}$ mice exceeded those observed in cKO/*Igh*$^{WT}$ mice (Supplementary Fig. 4b, c). These findings suggest that the genetic pre-arrangement of the *Igh* locus facilitates early to late LPre-B cell differentiation, even in the absence of UPF1.

## Commonalities and distinctions in the gene expression profiles of B cell progenitors lacking RBPs

Previous studies have demonstrated the critical role of RBPs, specifically ZFP36L1/L2 and CNOT3, in B cell development[7,10]. These studies showed that *Zfp36l1*$^{Flox/Flox}$ *Zfp36l2*$^{Flox/Flox}$ *Mb1*$^{cre/+}$ (*Zfp36l1/l2*-double conditional knockout (DCKO)) mice and *Cnot3*$^{Flox/Flox}$ *Mb1*$^{cre/+}$ (*Cnot3*-cKO) mice exhibited B cell differentiation defects just prior to V$_H$-D$_H$J$_H$ recombination at the *Igh* locus, a phenotype quite similar to that of *Upf1*-cKO mice. Consequently, we sought to compare the gene expression profiles of B cell progenitors lacking *Upf1*, *Zfp36l1/l2*, and *Cnot3*. The genes with low expression in *Upf1*-deficient early LPre-B cells showed enrichment in genes with low expression in *Zfp36l1/l2*-DCKO and *Cnot3*-cKO B cells (Fig. 5a, blue lines). In contrast, the highly expressed genes in *Upf1*-deficient early LPre-B cells exhibited enrichment solely in genes highly expressed in *Zfp36l1/l2*-DCKO B cells rather than in *Cnot3*-cKO B cells (Fig. 5a, red lines). This suggests that the RNA expression profile in *Upf1*-cKO B cells is closer to that of *Zfp36l1/l2*-DCKO B cells than *Cnot3*-cKO B cells.

The compromised V$_H$-D$_H$J$_H$ recombination observed in *Zfp36l1/l2*-DCKO B cells is attributed to the loss of cell cycle quiescence, a critical step for promoting V(D)J recombination[7,26]. Therefore, we delved into the regulatory role of UPF1 in the cell cycle of early LPre-B cells. We conducted BrdU assays to analyze the cell cycle status of B cells undergoing V$_H$-D$_H$J$_H$ recombination (CD19$^+$, B220$^+$, sIgM$^-$, CD43$^+$, CD2$^-$). The results revealed an increase in the number of cells in the S-phase among *Upf1*-deficient cells (Fig. 5b and c). In our RNA-seq analysis, E2F target genes and other cell cycle-related genes showed lower expression in *Upf1*-deficient early LPre B cells (Fig. 2b and i), probably because *Upf1*-cKO early LPre B cells lack pre-BCR, which induces potent cell cycles[2,27]. Nevertheless, even in the absence of pre-BCR expression, a set of cell cycle-related genes exhibited higher expression in *Upf1*-deficient early LPre B cells (Fig. 5d), implying that UPF1 is involved in suppressing some cell cycle-related genes. These results suggest that *Upf1*-deficiency promotes the transition from the G0/G1-phase to the S-phase, and the loss of cell cycle quiescence may emerge as a plausible cause for the hindrance of V$_H$-D$_H$J$_H$ recombination in *Upf1*-deficient early LPre-B cells.

## UPF1 is critical for sPre-Immature B cell differentiation

Although the expression of pre-rearranged *Igh* gene rescued early LPre- to late LPre-B cell differentiation in *Upf1*-cKO/*Igh*$^{B1-8hi}$ mice, the population of Imm- and Rec-B cells in the BM and splenic B cells was still diminished by *Upf1* depletion, indicating that UPF1 is also indispensable for B cell differentiation from sPre- to Immature B cell differentiation (Fig. 4i and Supplementary Figs. 4, 5). Since one of the major events during the sPre-B stage is the recombination at *Ig light chain* locus, we investigated whether the differentiation arrest observed in sPre-B cells is attributable to a deficiency in V$_L$-J$_L$ recombination at the *Ig light chain* locus during the sPre-B stage (Fig. 1a). The PCR assay revealed that V$_L$-J$_L$ recombination was not abrogated in cKO/*Igh*$^{B1-8hi}$ sPre-B cells (Fig. 6a), despite the concurrent absence of UPF1 protein expression in these cells (Fig. 6b). These findings indicate that V$_L$-J$_L$ recombination during the sPre-B stage can transpire in the absence of UPF1 in cKO/*Igh*$^{B1-8hi}$ mice.

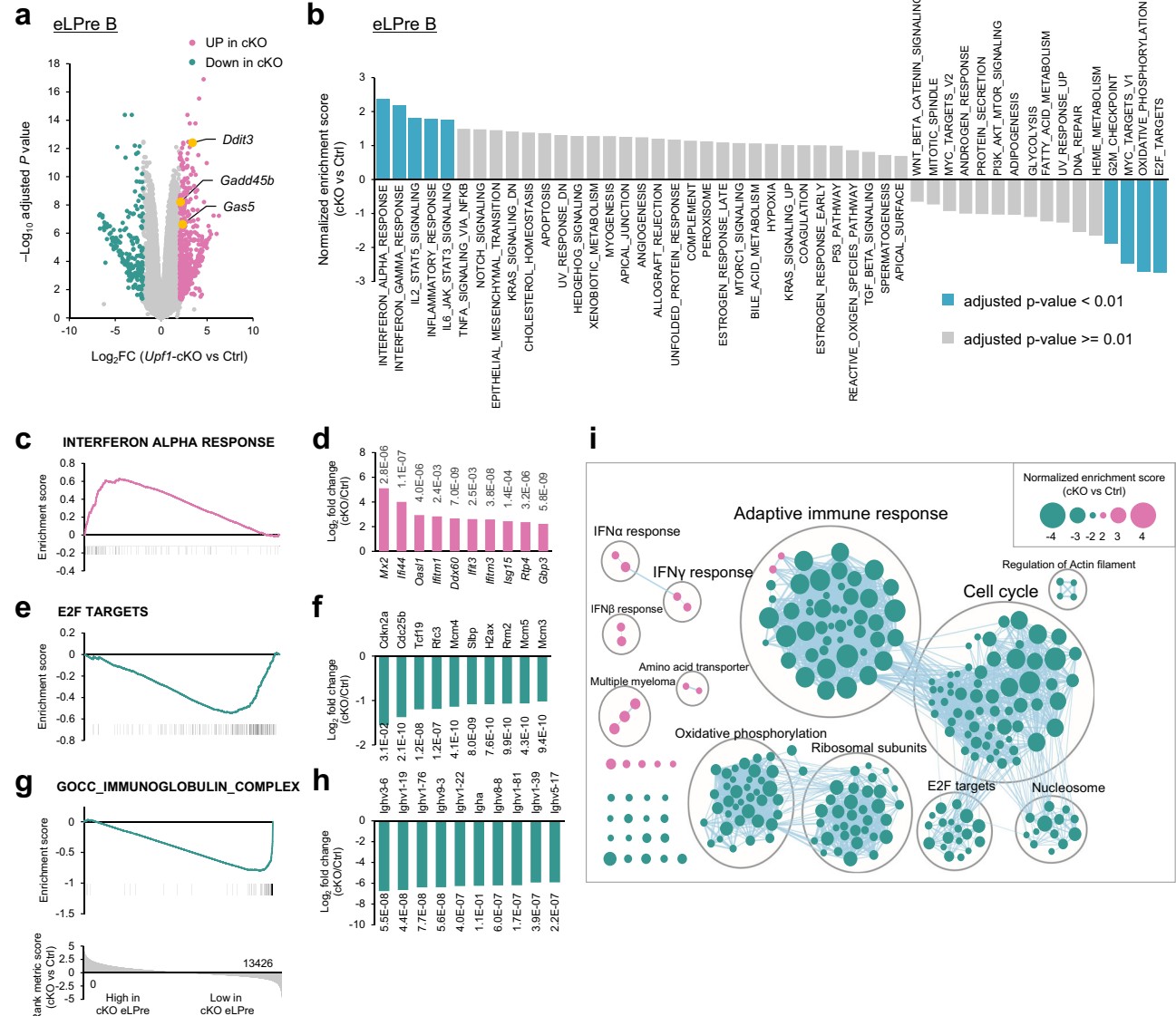

**Fig. 2 | Transcriptome analysis of Upf1-depleted early LPre-B cell. a** Volcano plot of the genes identified in the RNA-seq analysis of *Upf1*-cKO and Ctrl early LPre-B cells. Pink dots indicate highly expressed genes in cKO (Log2FC > = 2, adj. *P* < 0.05), and green dots indicate low expressed genes in cKO (Log2FC < = − 2, adj. *P* < 0.05). Known NMD targets (*Gas5*, *Gadd45b*, and *Ddit3*) are highlighted in yellow. The statistical analyses were performed using limma. Source data, including exact adjusted *p*-values, are provided as a Source Data file. **b** Normalized enrichment score of GSEA using hallmark gene sets. A positive number indicates the gene sets enriched in highly expressed genes in *Upf1*-cKO early LPre-B cells, and a negative number indicates those enriched in low expressed genes in *Upf1*-cKO. Significantly enriched gene sets (adj.*P* < 0.01) are colored in light blue. The statistical analyses

were performed using GSEA. **c–h** Enrichment score plots of "INTERFERON ALPHA RESPONSE" (**c**), "E2F TARGES" (**e**), and "IMMUNOGLOBULIN COMPLEX" (**g**) and bar graph showing Log2FC (*Upf1*-cKO vs Ctrl) of the representative genes in "INTER-FERON ALPHA RESPONSE" (**d**), "E2F TARGETS" (**f**), and "IMMUNOGLOBULIN COMPLEX" (H). The top 10 genes showing the highest (**d**) or lowest (**f, h**) Log2FC are picked up, and the number adjacent to each bar indicates adj.*P*. The statistical analyses were performed using limma. **i** The result of GSEA using all gene sets. Each node indicates identified gene sets and overlapping gene sets are grouped. The size of each dot indicates a normalized enrichment score. Pink and green nodes indicate the gene sets enriched in highly expressed genes and low expressed genes in cKO, respectively.

To further investigate the mechanisms by which UPF1 controls later phases of B cell development in the presence of the pre-rearranged *Igh* allele, we conducted RNA-seq analysis using early LPre- and sPre-B cells derived from cKO/*Igh*[B1-8hi] and Ctrl/*Igh*[B1-8hi] mice (Supplementary Data 2). By comparing transcriptomic profiles between early LPre- and sPre-B cells from Ctrl/*Igh*[B1-8hi] (control) mice, we noted dynamic changes in gene expression during the transition from early LPre- to sPre-B cells: the downregulation of 1533 genes (Log2FC < = − 2, adjP < 0.05) and upregulation of 847 genes (Log2FC > = 2, adjP < 0.05) (Fig. 6c). Interestingly, the genes that showed decreased expression levels during the differentiation from early LPre- to sPre-B cells in

control cells exhibited higher expression in *Upf1*-cKO/*Igh*[B1-8hi] sPre-B cells than Ctrl/*Igh*[B1-8hi] sPre-B cells (Fig. 6d).

Further GSEA revealed the enrichment of gene sets associated with cell cycle progression, such as E2F TARGETS (adj. *P* = 0.087), as well as UNFOLDED PROTEIN RESPONSE (UPR) (adj. *P* = 0.37, *P* = 0.021) in *Upf1*-cKO/*Igh*[B1-8hi] sPre-B cells compared to Ctrl/*Igh*[B1-8hi] cells (Fig. 6e). We next investigated transcriptome transition from early LPre- to sPre-B cells in Ctrl/B1-8[hi] and cKO/B1-8[hi] (Supplementary Fig. 6a, b). In control mice, during the differentiation from early LPre- to sPre-B cells, the gene sets related to cell cycling, such as the E2F target, G2M_Chcekpoint, and MYC target, exhibited downregulation

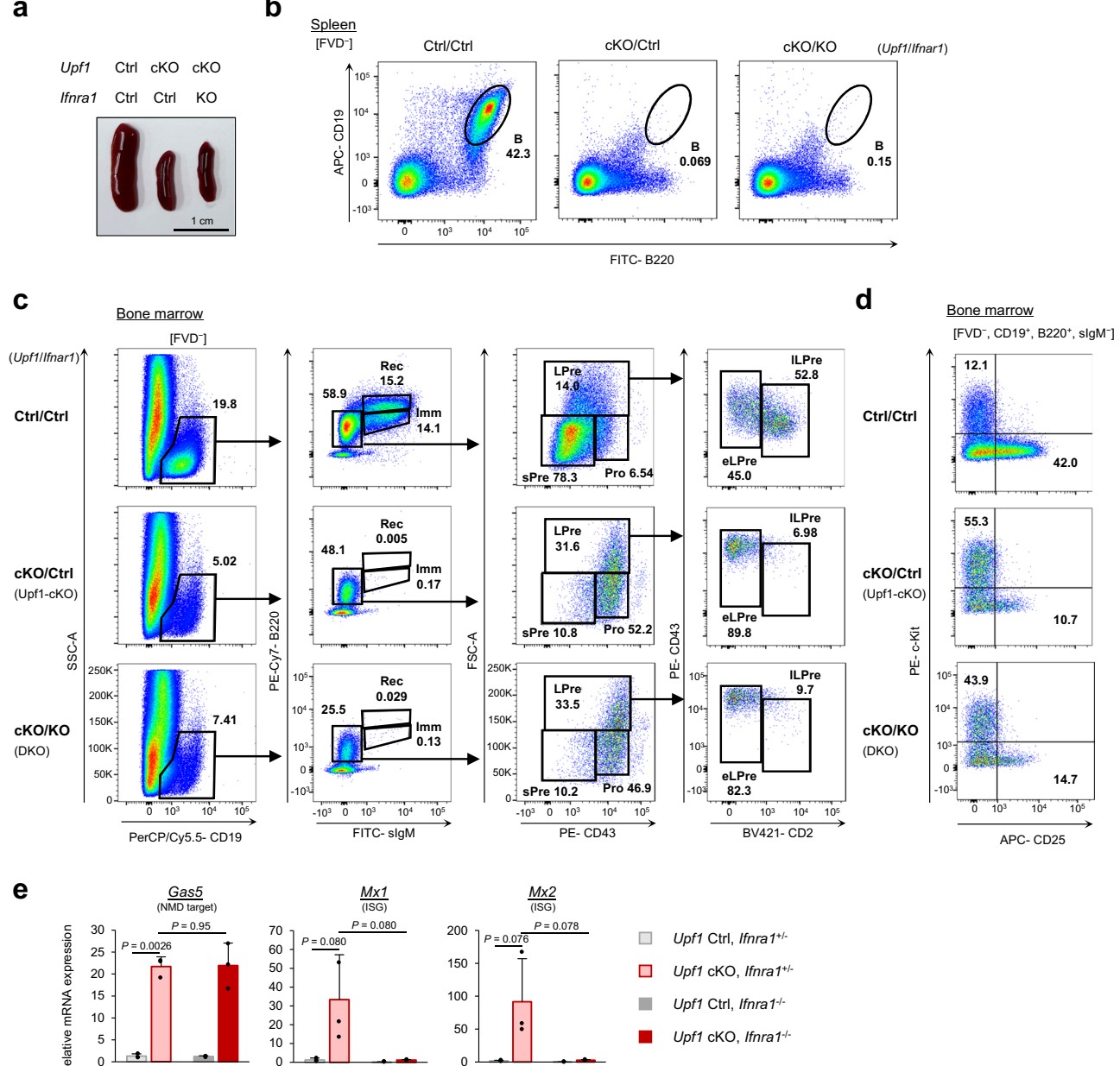

**Fig. 3 | Blocking of IFN receptor signal does not rescue the B cell development in *Upf1*-cKO mice. a** Appearance of mouse spleens from indicated genotypes. Results are representative of at least three independent experiments. **b** Flow cytometry plots of indicated populations in the splenocytes derived from indicated mice. Results are representative of at least two independent experiments. **c** Flow cytometry plots of indicated populations in the BM of indicated mice. **d** Flow cytometry plots of FVD⁻, CD19⁺, B220⁺, sIgM⁻ cells derived from the BM of indicated mice. Results are representative of at least two independent experiments. **e** The mRNA expressions of indicated genes in early LPre-B cells were derived from the indicated mice ($n = 3$ per group). Each bar represents the mean ± SD from biological replicates. Statistical significance was calculated with a two-sided Student's *t* test. Source data are provided as a Source Data file.

(Supplementary Fig. 6a). These alterations align with the transition from actively proliferating large Pre-B cells to quiescent small Pre-B cells. The enrichment of the E2F target gene set in *Upf1*-deficient sPre-B cells (cKO/*Igh*^B1-8hi), but not early LPre-B cells (Fig. 6e), suggests a potential requirement of UPF1 in driving the downregulation of cell cycle-related gene expression during the transition from early LPre- to sPre-B stage.

In contrast, consistent with the results of PCR assay for the *Ig light chain*, the expression levels of v-region transcripts from *Ig light chains* (*Igkv* and *Iglv*) were largely similar between UPF1- cKO/*Igh*^B1-8hi and Ctrl/*Igh*^B1-8hi sPre-B cells (Fig. 6f). This finding suggests that UPF1 is dispensable for the recombination of *Ig light chain* DNA.

Notably, the gene sets "INTERFERON ALPHA RESPONSE" and "INTERFERON GAMMA RESPONSE", which were significantly enriched in cKO early LPre-B cells, did not show enrichment in early LPre-B cells in the absence of UPF1 with the pre-rearranged *Igh* allele (cKO/*Igh*^B1-8hi) (Fig. 6g). Consistently, genes related to the IFN response were only modestly elevated in cKO/*Igh*^B1-8hi early LPre-B cells, whereas these genes were more robustly increased in cKO/*Igh*^WT early LPre-B cells (Fig. 6h). These data suggest that the *Igh* pre-rearrangement prevented the aberrant expression of IFN response-related genes in *Upf1*-deficient early LPre-B cells. In contrast, IFN response-related genes were upregulated in sPre-B cells from cKO/*Igh*^B1-8hi compared with Ctrl/*Igh*^B1-8hi mice (Fig. 6h). For instance, ISGs, such as *Ifi44* and *Ddx60*, started to

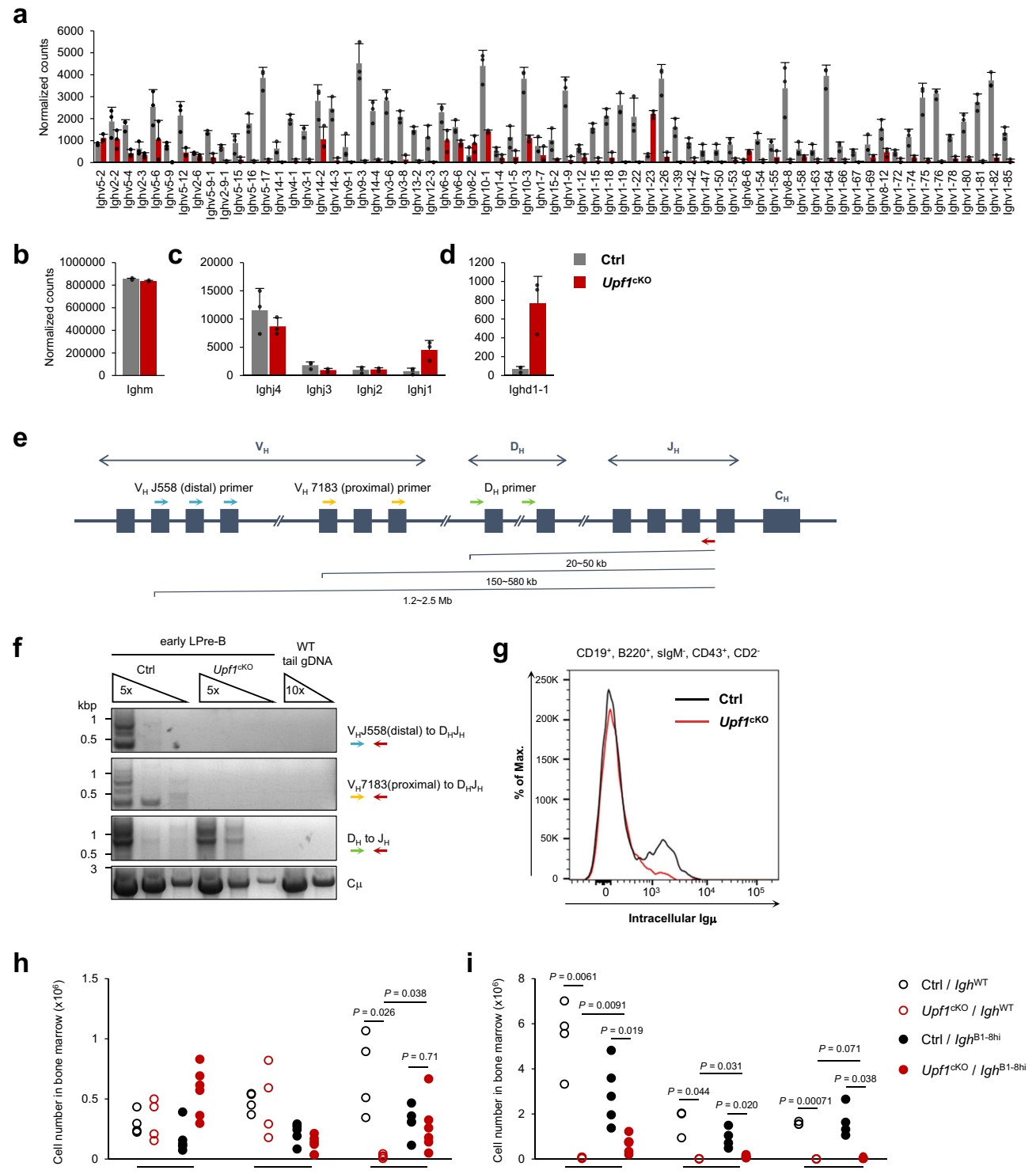

**Fig. 4 | UPF1 is necessary for Igh $V_H$-$D_H J_H$ recombination at early LPre-B stage.**
**a**–**d** The transcript amount of *Ighv* (**a**), *Ighm* (**b**), *Ighj* (**c**), and *Ighd* (**d**) in *Upf1*-cKO and Ctrl early LPre-B cells. The data were obtained from the RNA-seq shown in Fig. 2 (*n* = 3). Each bar represents the mean ± SD from biological replicates. **e** Schematic illustration of germline *Igh* locus. Colored arrows indicate each primer used in PCR analysis in (**f**). **f** PCR assay amplifying recombined *Igh* as shown in (**e**) using the DNA derived from *Upf1*-cKO and Ctrl early LPre-B cells. Genome DNA derived from the tail of the WT mouse was used as a negative control. Results are representative of at

least three independent experiments. **g** The histogram of intracellular staining of Igμ chain in CD19+, B220+, sIgM-, CD43+, CD2- cells derived from indicated mice. Results are representative of two independent experiments. **h**, **i** Cell numbers of indicated populations in the BM from indicated mouse genotypes (Pro-, early LPre-, late LPre-, and sPre-B; *N* = 4 (Ctrl/*Igh*$^{WT}$, cKO/*Igh*$^{WT}$), *N* = 5 (Ctrl/*Igh*$^{B1-8hi}$), *N* = 6 (cKO/ *Igh*$^{B1-8hi}$). Imm- and Rec-B; *N* = 3 (Ctrl/*Igh*$^{WT}$, cKO/*Igh*$^{WT}$), *N* = 4 (Ctrl/*Igh*$^{B1-8hi}$), *N* = 5 (cKO/*Igh*$^{B1-8hi}$)). Statistical significance was calculated with a two-sided Student's *t* test. Source data are provided as a Source Data file.

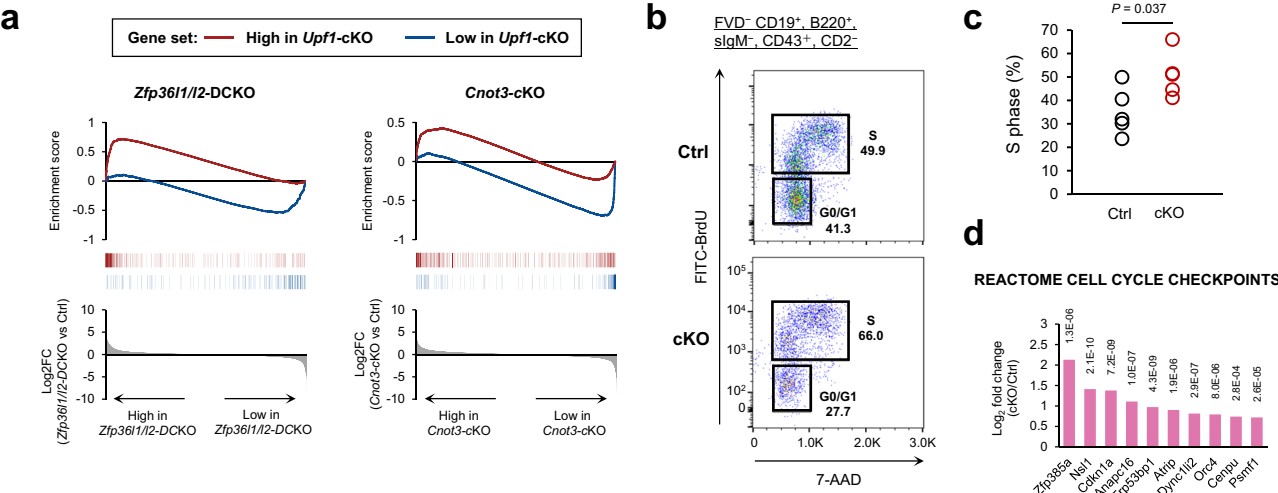

**Fig. 5 | Phenotypic similarity between Upf1-cKO mice and Zfp36l1/l2-DCKO mice. a** Enrichment score plot of the differentially expressed genes of *Upf1*-deficient early LPre-B cells in *Zfp36l1/l2*-DCKO or *Cnot3*-cKO progenitor B cells. The red line indicates enrichment of the high expression genes in *Upf1*-cKO cells (Log2FC > = 2, adjP < 0.05), and the blue line indicates enrichment of the low expression genes in *Upf1*-cKO cells (Log2FC < = − 2, adjP <0.05). **b, c** The representative flow cytometry plots of BrdU assay in FVD⁻ CD19⁺, B220ᵐⁱᵈ, sIgM⁻, CD43⁺,

CD2⁻ cells (**c**) and the percentages of cells in S-phase (*N* = 5 for each genotype). Statistical significance was calculated with a two-sided Student's *t* test. **d** Highly expressed genes included in the "REACTOME CELL CYCLE CHECKPOINTS" gene set in *Upf1*-cKO early LPre-B cells. Bars indicate Log2FC (*Upf1*-cKO vs Ctrl), and the number adjacent to each bar indicates adj.*P*. The statistical analyses were performed using limma. Source data are provided as a Source Data file.

elevate in cKO/*Igh*^B1-8hi^ sPre-B cells (Supplementary Fig. 6c), suggesting that the *Igh* pre-rearrangement failed to suppress the expression of IFN response-related genes in sPre-B cells lacking UPF1. These results imply that UPF1 plays a role in preventing the abnormal expression of IFN response-related genes not only in the early LPre-B stage but also in the sPre-B stage.

The UPR acts as a critical checkpoint in B cell development by regulating the maintenance of BCR expression and proper transport to the cell surface[28–30]. Genes associated with the UPR were highly enriched in both early LPre-B (adj. *P* = 0.0075, *P* < 0.0001) and sPre-B cells (adj. *P* = 0.37, *P* = 0.021) from cKO/*Igh*^B1-8hi^ mice (Fig. 6e, g). We also found a comparable number of UPR genes affected by UPF1 depletion in *Igh*^WT^ early LPre-, *Igh*^B1-8hi^ early LPre-, and *Igh*^B1-8hi^ sPre-B cells (Fig. 6i). These data suggest that, unlike IFN-related genes, the omission of *Igh* recombination through B1-8^hi^ knock-in did not mitigate the abnormally high expression of UPR genes in *Upf1*-deficient B cells. Some of the UPR genes were downregulated during the differentiation from early LPre- to sPre-B stage in *Upf1*-Ctrl/*Igh*^B1-8hi^ B cells (GSEA: adj.*P* = 0.060) (Fig. 6j and Supplementary Fig. 6a), suggesting a delayed downregulation of UPR genes during the transition from early LPre- to sPre-B stage in *Upf1*-deficient B cells. Collectively, these data demonstrate that UPF1 is critical for ensuring the transcriptome shift from the early LPre- to sPre-B stage by preventing abnormal expression of genes related to cell cycle, IFN response, and the UPR.

## UPF1 downregulates the expression of its target RNAs in early LPre- and sPre-B stages

We next searched for RNAs directly targeted by UPF1 in B cells. It was reported that immunoprecipitation with anti-p-UPF1 (S1111) successfully isolated NMD-target mRNAs, while steady-state UPF1 largely nonspecifically associated with mRNAs[22]. We conducted RNA-immunoprecipitation sequencing (RIP-seq) analysis using anti-p-UPF1 antibody in activated splenic B cells since it was difficult to isolate a sufficient amount of early LPre-B cells for this experiment (Fig. 7a and Supplementary Fig. 7a). We identified 812 p-UPF1-bound RNAs, including prominent NMD targets such as *Smg5*, *Gadd45b*, and *Ddit3* (Fig. 7b and Supplementary Data 3). Gene ontology analysis revealed

various functions of UPF1-target mRNAs, such as NF-κB activation and the UPR (Fig. 7c).

Interestingly, we found that the RNAs directly targeted by p-UPF1 were enriched in *Upf1*-cKO early LPre-B, cKO/*Igh*^B1-8hi^ early LPre-B cells, and cKO/*Igh*^B1-8hi^ sPre-B cells (Fig. 7d–f). These data suggest that UPF1 directly downregulates the expression of the target mRNAs in both early LPre- and sPre-B stages. On the other hand, the RNAs identified as UPF1-targets were not enriched in *Cnot3*-deficient or *Zfp36l1/l2*-deficient progenitor B-cells, indicating that not all UPF1-binding RNAs are concurrently regulated by ZFP36L1/L2 or CNOT3 despite the transcriptome similarity between *Upf1*-deficient early LPre- and *Zfp36l1/l2*-deficient progenitor B cells (Supplementary Fig. 7b, c). These results suggest that although the initial dysregulations of RNA metabolism differ, *Upf1*-cKO and *Zfp36l1/l2*-DCKO B cells eventually converge toward similar RNA expression profiles.

Since our RIP-seq analysis did not cover the entire transcripts in early LPre- and sPre-B cells, identifying UPF1-target RNAs among all the highly expressed genes in *Upf1*-deficient progenitor B cells posed a challenge (Supplementary Fig. 7d). However, we observed that UPR genes targeted by UPF1 were highly expressed in *Upf1*-deficient early LPre- and sPre-B cells (Fig. 7g). In addition, UPF1-targeted mRNAs in IFN response- or cell cycle-related genes were also highly expressed in *Upf1*-deficient early LPre-B cells (Fig. 7h, i). These findings suggest that UPF1-mediated direct mRNA suppression contributes to the appropriate gene expression status at the early LPre- and sPre-B stages of B cell development.

## Discussion

In this study, we have elucidated the vital role of UPF1 in B cell development. The disruption in B cell development observed in the absence of UPF1 is directly attributed to the impediment of *Igh* V$_H$-D$_H$J$_H$ recombination during the early LPre-B stage. The association established between UPF1 and V$_H$-D$_H$J$_H$ recombination elucidates a fundamental mechanism, underscoring the indispensable role of UPF1 in orchestrating B cell differentiation from early LPre- to late LPre-B stage through V$_H$-D$_H$J$_H$ recombination in the mouse BM. In addition, UPF1 is essential for sPre-immature B cell differentiation. UPF1 suppresses a

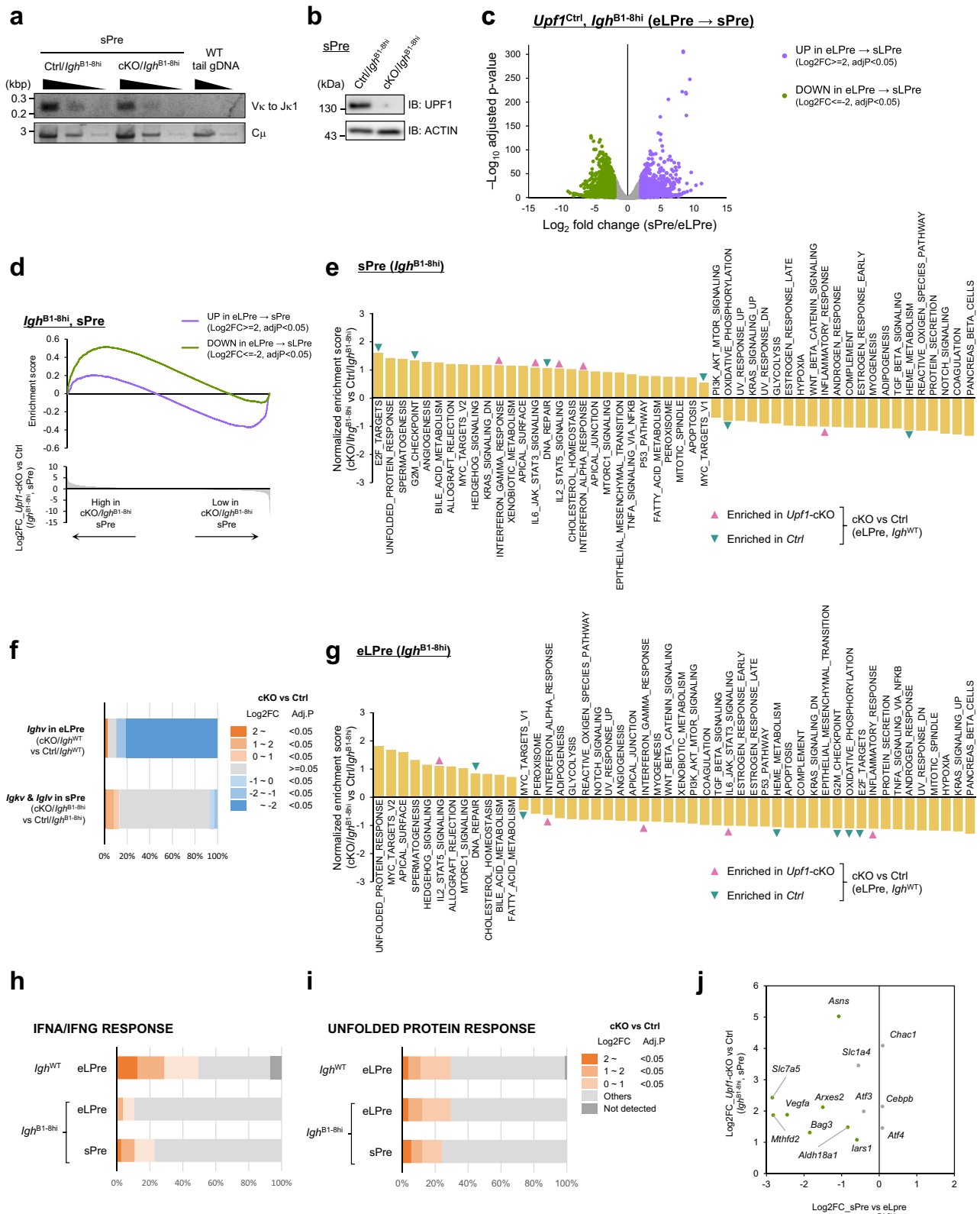

specific set of RNAs associated with cell cycle regulation, the UPR, and IFN response during B cell differentiation.

Remarkably, we observed a significant upregulation in the protein expression and phosphorylation of UPF1 during the early LPre-B cell stage, coinciding with the occurrence of $V_H$-$D_H J_H$ recombination. Several explanations for this phenomenon are plausible. Firstly, it is conceivable that the translation efficiency of

*Upf1* mRNA is augmented in early LPre-B cells. Alternatively, the activity of SMG1 kinase, a key player in UPF1 phosphorylation, might be modulated during B cell differentiation. Subsequent investigations are warranted to elucidate whether specific signaling pathways exist to directly control the translation and phosphorylation of UPF1, thereby exerting precise control over B cell differentiation.

**Fig. 6 | UPF1 ensures transcriptome dynamics throughout the early LPre- to sPre-B cell differentiation. a** PCR assay amplifying recombined *Igk* in sPre-B cells derived from the indicated mice. Genome DNA derived from the tail of WT was used as a negative control. Results are representative of two independent experiments. **b** Immunoblot showing protein expression of UPF1 in sPre-B cells from Ctrl/*Igh*^B1-8hi and cKO/*Igh*^B1-8hi mice. Results are representative of at least three independent experiments. **c** Volcano plot of RNA expression comparison between early LPre- and sPre-B cells (Ctrl/*Igh*^B1-8hi). The upregulated genes and the downregulated genes in sPre-B cells from early LPre-B cells are colored purple and green, respectively. The statistical analyses were performed using edgeR. **d** Enrichment of the upregulated genes in sPre-B cells (purple) and the downregulated genes in sPre-B cells (green) in *Upf1*-cKO/*Igh*^B1-8hi sPre-B cells. **e, g** Normalized enrichment score of GSEA (*Upf1*-cKO/*Igh*^B1-8hi vs Ctrl/*Igh*^B1-8hi) in sPre-B cells (**e**) and early LPre-B cells (**g**). The gene sets positively or negatively enriched in *Upf1*-cKO/*Igh*^WT early LPre-B cells

(Fig. 2b) are pointed with pink or green arrows, respectively. **f** The ratio of differentially expressed transcripts derived from *Igh* v region (*Upf1*-cKO vs Ctrl in early LPre-B) and *Igk*/*Igl* v region (*Upf1*-cKO/*Igh*^B1-8hi vs Ctrl/*Igh*^B1-8hi in sPre). The statistical analyses were performed using limma (*Igh*^WT), and edgeR (*Igh*^B1-8hi). **h, i** The ratio of highly expressed IFNA/IFNG RESPONSE genes (**h**) and UNFOLDED PROTEIN RESPONSE genes (**i**) in *Upf1*-deficient B cells with indicated genotypes and cell types. Hallmark gene sets from GSEA are used (IFN ALPHA RESPONSE and IFN GAMMA RESPONSE are combined without gene overlap). The statistical analyses were performed using limma (*Igh*^WT), and edgeR (*Igh*^B1-8hi). **j** Comparison of UPR-related gene expression between early LPre- and sPre-B cells (Ctrl/*Igh*^B1-8hi). Green dots indicate significantly upregulated genes in sPre-B cells compared to early LPre-B cells. Only the genes highly expressed in *Upf1*-cKO/*Igh*^B1-8hi sPre-B cells are shown. Source data are provided as a Source Data file.

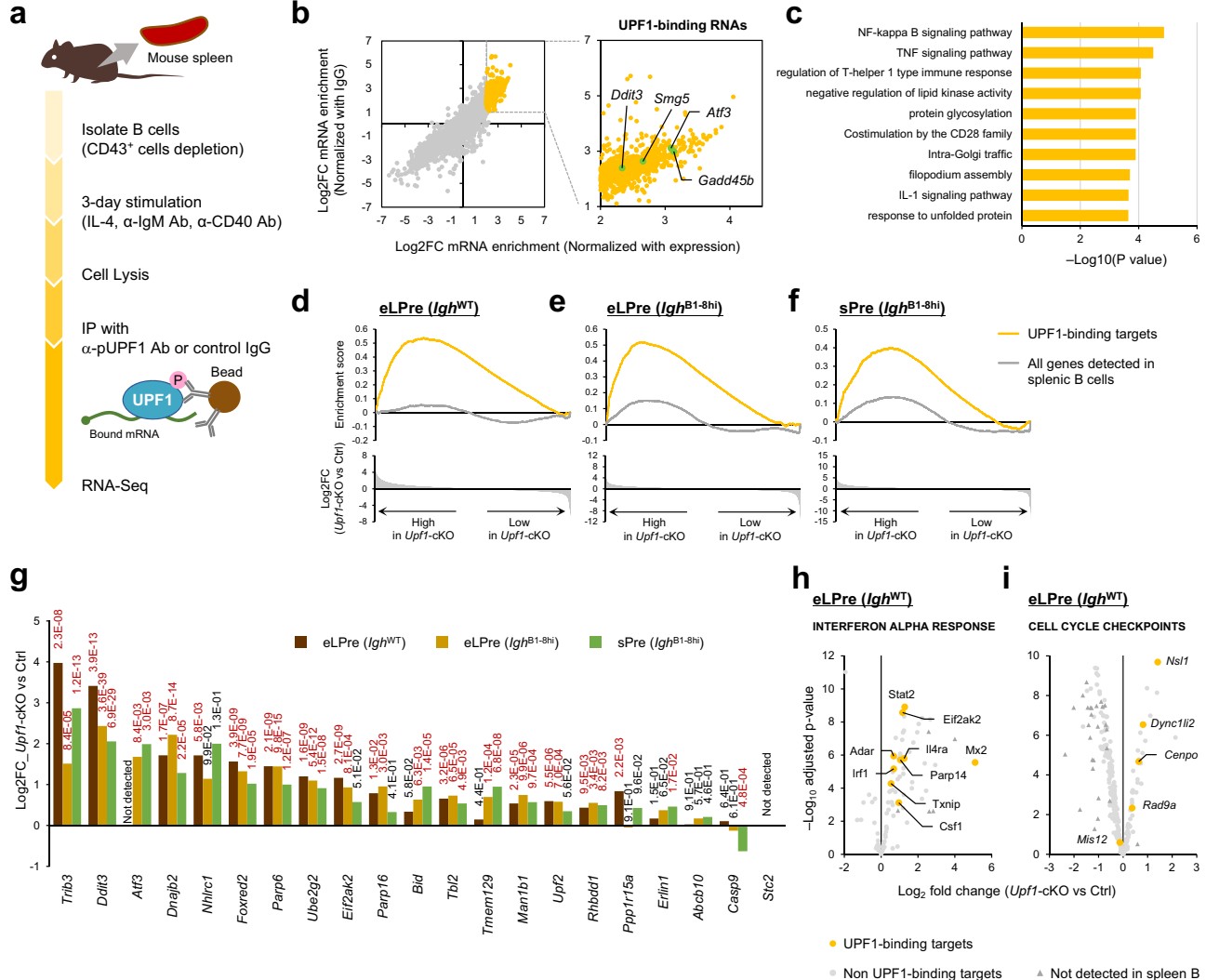

**Fig. 7 | RIP-Seq analysis to determine the direct binding targets of p-UPF1. a** Scheme used to identify UPF1 target RNAs in B cells with RIP-seq. Isolated splenic B cells were stimulated for 3 days. After stimulation, the B cells were lysed in RNA IP lysis buffer and incubated with anti-phospho-UPF1 Ab bound to protein G magnetic beads. The precipitated RNAs were eluted from the beads and analyzed by RNA-seq. **b** RNAs precipitated with p-UPF1 antibody in splenic B cells. The *x*-axis indicates Log2FC normalized with RNA expression in splenic B cells (input RNAs) and the *y*-axis indicates Log2FC normalized with RNAs precipitated with control IgG antibody. Yellow dots indicate highly enriched RNAs; we defined them as UPF1-binding targets (see "Methods"). Known NMD targets are highlighted in light green. **c** Gene ontology analysis of UPF1-target RNAs. The statistical analyses were

performed using Metascape. **d–f** Enrichment score plot of UPF1-binding targets (yellow) and all genes detected in splenic B cells (gray) in *Upf1*-cKO vs Ctrl (**d**); early LPre-B, *Igh*^WT, (**e**); early LPre-B, *Igh*^B1-8hi, (**f**); sPre-B, *Igh*^B1-8hi (**f**). **g** Comparison of "response to unfolded protein expression (GO:0006986)" gene expression between *Upf1*-cKO vs Ctrl in indicated genotypes and cell types. The number adjacent to each bar indicates adj.*P* (red; adj.*P* < 0.05). Only the genes identified as UPF1-target are shown. The statistical analyses were performed using limma (*Igh*^WT), and edgeR (*Igh*^B1-8hi). **h, i** Volcano plot comparing *Upf1*-cKO and Ctrl (early LPre-B) of the genes included in INTERFERON ALPHA RESPONSE (**h**) and CELL CYCLE CHECKPOINTS (**i**). Yellow dots indicate the genes identified as UPF1-targets. The statistical analyses were performed using limma.

An intriguing question that remains to be addressed is the precise mechanism by which UPF1 orchestrates the regulation of $V_H$-$D_H J_H$ recombination in early LPre-B cells. We observed that the loss of cell cycle quiescence in *Upf1*-deficient early LPre-B cells is one potential factor contributing to the inhibition of $V_H$-$D_H J_H$ recombination. Moreover, previous studies have provided evidence that germline gene transcripts (GLTs) of *Igh* play a supportive role in DNA recombination[31,32]. Notably, the expression of transcripts originating from the V region of *Igh* was significantly reduced in *Upf1*-deficient early LPre-B cells. In contrast, $D_H$-$J_H$ recombination, a process preceding $V_H$-$D_H J_H$ recombination, proceeded even in *Upf1*-deficient B cells, with the levels of transcripts arising from the D or J region of *Igh* remaining unaltered in *Upf1*-deficient early LPre-B cells. Therefore, this suggests that UPF1 might exert control over the expression of GLTs of the *Igh* V region, potentially initiating $V_H$-$D_H J_H$ recombination. Further studies are required to shed light on the intricate mechanisms governing how UPF1 regulates $V_H$-$D_H J_H$ recombination.

Through transcriptome analysis, we have identified a spectrum of genes that exhibit differential expression in the *Upf1*-deficient early LPre-B cells. Among these changes, one of the most striking alterations was the marked increase in the expression of ISGs. The abnormally heightened expression of ISGs was attenuated upon the knockout of the IFN receptor, implying the activation of the IFN signaling pathway in *Upf1*-deficient early LPre-B cells. Interestingly, pre-rearrangement of *Igh*, conducted by crossing *Igh*[B1-8hi] knock-in mice, also inhibited the abnormally high expression of IFN-response genes in *Upf1*-deficient early LPre-B cells. On the other hand, *Upf1*-cKO/*Igh*[B1-8hi] sPre-B cells, in which $V_L$-$J_L$ recombination occurs, showed higher expression of IFN-response genes. Given that both *Upf1*-cKO early LPre-B cells and *Upf1*-cKO/*Igh*[B1-8hi] sPre-B cells undergo DNA recombination at $D_H$-$J_H$ and $V_L$-$J_L$ regions, respectively, it is tempting to hypothesize that DNA recombination triggered the aberrant activation of IFN pathway in *Upf1*-deficient B cells. Despite this IFN signaling activation, the B cell developmental blockade was not rectified in *Upf1/Ifnar1* double knockout mice. These results suggest that the accumulation of ISGs due to *Upf1* deficiency does not significantly influence the transition from early to late LPre-B cells. It is well documented that spontaneous expression of ISGs can be induced by the abnormal accumulation of double-stranded (ds) RNA, often stemming from disruptions caused by the failure of RNA metabolisms, such as the loss of METTL3, an $N^6$-methyladenosine (m$^6$A) methyltransferase[33], or the absence of an RNA editing enzyme ADAR1[34]. Given that impaired NMD due to *Upf1* deficiency may lead to the accumulation of aberrant mRNAs, it is also possible that UPF1 plays a critical role in suppressing IFN responses by preventing aberrant buildup of dsRNA.

Previous studies have reported that the deficiency of certain other RBPs leads to a halt in B cell differentiation, typically occurring just before the $V_H$-$D_H J_H$ recombination[4,7,8,10–12,35]. Among these RBPs, CNOT3 is a component of the CCR4–NOT deadenylase complex, responsible for promoting poly(A) tail shortening in various mRNA decay machinery, including NMD and ZFP36L1/2-mediated mRNA decay. However, our findings indicate that the RNA expression profile of early LPre-B cells in *Upf1*-cKO mice is closer to that of *Zfp36l1/l2*-DCKO B cells than *Cnot3*-cKO B cells. We must mention that the slight disparities in data acquisition methods among these three data sets, such as variations in the gating strategy for early developing B cells, might impact the analysis results. Furthermore, our investigation revealed that the expression of p-UPF1-binding RNAs was not enriched in either *Zfp36L1/L2*-DCKO or *Cnot3*-cKO B cells. In the context of the NMD pathway, RNA degradation is mediated by the NMD-specific endonuclease SMG6, alongside the SMG5 and SMG7 proteins, which recruit the CCR4–NOT deadenylase complex[15,18]. SMG6, under the guidance of UPF1, triggers endonucleolytic cleavage of mRNAs, independent of deadenylation[16,17]. Thus, the SMG6-mediated RNA degradation pathway may exhibit a preference for early LPre-B cells.

An unresolved question pertains to how UPF1 regulates the differentiation from sPre-B cells to immature B cells, a process involving *Ig light chain* recombination and subsequent expression of BCR (IgM) on the cell surface. While *Upf1*-deficient sPre-B cells were capable of undergoing *Ig light chain* recombination, they failed to express BCR on the cell surface. Our investigation revealed elevated expression of UPR genes, some of which are directly targeted by UPF1, in *Upf1*-deficient sPre-B cells. The increased expression of UPR genes in *Upf1*-deficient cells might exacerbate unnecessary UPR, potentially impeding the expression of BCR and its proper transport to the cell surface[28–30,36]. The disruption could hinder the differentiation process from sPre-B cells to immature B cells.

We found that UPF1 directly suppresses a specific set of RNAs associated with cell cycle regulation, the UPR, and IFN responses during B cell differentiation. However, it remains unclear whether UPF1 targets 40 to 60% of highly expressed genes in *Upf1*-deficient progenitor B cells, as the RNAs identified in RIP-seq using splenic B cells did not encompass all the genes expressed in early LPre- or sPre-B cells. Therefore, there is a possibility of additional target mRNAs specifically expressed in early LPre- and sPre-B cells that are regulated by UPF1. Furthermore, considering the dynamic changes in the abundance of phosphorylated UPF1 during B cell differentiation, it is intriguing to investigate whether the binding specificity of UPF1 varies depending on the stages of B cell differentiation.

In summary, this study collectively underscores the intrinsic role of UPF1 as a pivotal factor in the transition from early to late LPre-B and sPre-B to Immature-B differentiation. UPF1 is prerequisite for $V_H$-$D_H J_H$ recombination without affecting $D_H$-$J_H$ recombination. Furthermore, UPF1 ensures the transcriptome dynamics from early LPre- to sPre-B cells. Our findings contribute to a deeper understanding of the intricate molecular mechanisms governing B cell differentiation.

## Methods

### Study approval

All animal experiments were conducted in compliance with the regulations approved by the Committee for Animal Experiments of the Graduate School of Medicine and Institute for Frontier Life and Medical Sciences, Kyoto University.

### Mice

The targeting vector was constructed by inserting a 2.0 kb fragment containing exons 4 to 6 of the *Upf1* gene flanked by loxP sites, 8.0 kb of a 5′ sequence, 1.0 kb of a 3′ sequence, and a neomycin (neo)-resistant gene flanked by FRT sites into a PGKneoF2L2DTA vector. The targeting vector was linearized and transfected into EGR-101 embryonic stem (ES) cells derived from C57BL/6 mice by electroporation, and G418-resistant clones were screened for homologous recombination by PCR and Southern blot analysis. The primers used for PCR are listed in Supplementary Table 1. The successfully recombined clones were microinjected into blastocytes derived from ICR mice and transferred to pseudopregnant females. Mating of chimeric male mice to C57BL/6 female mice resulted in the transmission of the floxed allele to the germline. The *Upf1*[Flox/+] mice were crossed with FLP-Cre mice (C57BL/6 background) to eliminate the neo gene and then crossed with the indicated Cre expressing mice (C57BL/6 background).

Mb1[cre/+] mice (C57BL/6 background)[37] were kindly provided by Dr. Michael Reth (University of Freiburg). $V_H$B1-8[high] knock-in mice (termed in this study as B1-8[hi]) (C57BL/6 background)[25] were kindly provided by Dr. Michel C. Nussenzweig (The Rockefeller University). Mice were used at 8 to 16 weeks old, male and female, and were age-matched. Mice were euthanized by cervical dislocation. Mice were co-housed with the control mice and maintained under specific pathogen-free (SPF) conditions at Kyoto University animal facility with a 12 h light/dark cycle and access to food and water ad libitum. Room temperature was maintained at 23 ± 3 °C, with a relative humidity of 50 ± 20% and all

experiments were performed in accordance with institutional guidelines at Kyoto University.

## Immunoblot analysis

Whole-cell extracts were prepared in lysis buffer (1% (vol/vol) Nonidet P-40, 0.1% (wt/vol) SDS, 1% (wt/vol) sodium deoxycholate, 150 mM NaCl, and 20 mM Tris-HCl, pH 8.0) with PhosSTOP (Sigma) and Protease Inhibitor Cocktail (nacalai tesque) and suspended in SDS sample buffer (50 mM Tris-HCl, pH 6.8, 2% (wt/vol) SDS, 5% (vol/vol) β-mercaptoethanol, 10% (vol/vol) glycerol and bromophenol blue). Proteins were boiled for 5 min at 95 °C. Lysates from the equivalent number of cells were resolved on polyacrylamide gels (e-PAGEL; ATTO) and transferred onto 0.2 μm pore size Immun-Blot PVDF membranes (Bio-Rad). Membranes were incubated with indicated primary antibodies and HRP-coupled secondary antibodies (NA9340; GE Healthcare). The following primary antibodies were used for immunoblot analysis: antibody to UPF1 (NBP1-05967, Novus biologicals), Phospho-UPF1(07-1016, Merck), β-actin (sc-1615, Santa Cruz) The complete list of antibodies used in this study can be found in Supplementary table 2.

## Flow cytometry analysis and isolation of B cell progenitors

Single-cell suspension of splenocytes or BM cells were stained with Fixable Viability Dye eFluor 450, 506, or 780 (65-0863-14, 65-0866-14, or 65-0865-14, respectively, eBioscience) for 30 min at 4 °C and indicated antibody for 30 min at 4 °C, followed by the staining with fluorescent-labeled streptavidin (405225, Biolegend) for 30 min. The cells were washed twice and suspended with FACS buffer (0.5 % BSA and 2 mM EDTA in PBS). Antibodies and fluorescent dyes for flow cytometry analysis were as follows: Biotin anti-mouse CD2 (Biolegend, 100103, clone: RM2-5), PerCP/Cy5.5 anti-mouse CD19 (Biolegend, 152405, clone: 1D3), FITC anti-mouse CD19 (Biolegend, 115506, clone: 6D5), Alexa Fluor 700 anti-mouse CD19 (Biolegend, 115527, clone: 6D5), APC Rat Anti-Mouse CD19 (BD Pharmingen, 550992, clone: 1D3), APC anti-mouse CD25 (Biolegend, 102011, clone: PC61), PE Rat Anti-Mouse CD43 (BD Pharmingen, 561857, clone: S7), FITC anti-mouse IgM (Biolegend, 406505, clone: RMM-1), APC anti-mouse IgM (Biolegend, 406509, clone: RMM-1), PE-Cy7 Rat Anti-Mouse CD45R/B220 (BD Pharmingen, 552772, clone: RA3-6B2), FITC anti-mouse/human CD45R/B220 (Biolegend, 103206, clone: RA3-6B2), PE anti-mouse/human CD45R/B220 (Biolegend, 103208, clone: RA3-6B2), PE anti-mouse CD117 (c-Kit) (Biolegend, 105807, clone: 2B8), PerCP Hamster Anti-Mouse CD3e (BD Pharmingen, 553067, clone: 145-2C11), PerCP/Cy5.5 anti-mouse/human CD11b (Biolegend, 101228, clone: M1/70), PerCP anti-mouse CD11c (Biolegend, 117326, clone: N418), PerCP anti-mouse F4/80 (Biolegend, 123125, clone: BM8), PerCP/Cy5.5 anti-mouse TER-119 (Biolegend, 116227, clone: TER-119), Brilliant Violet 421 Streptavidin (Biolegend, 405225). The anti-CD3e, anti-CD11b, anti-CD11c, anti-F4/80, and anti-TER-119 antibodies were used for the lineage cell staining. For intracellular Igμ staining, BM cells were firstly stained for surface antigens, and then fixed, permeabilized with Intracellular Fixation & Permeabilization Buffer Set (eBioscience), and stained with anti-IgM antibody (406509 (APC), Biolegend) The antibodies used for the flow cytometry analysis are listed in Supplementary table 2. Data were collected using FACSVerse, FACSAriaII, FACSAriaIII or LSRFortessa (BD Biosciences), and analyzed using FlowJo (BD Biosciences).

The B cell progenitors were identified as Pre-pro-B (Lineage⁻, CD19⁻, B220⁺, sIgM⁻), Pro-B (Lineage⁻, CD19⁺, B220⁺, sIgM⁻, CD43$^{hi}$, FSC$^{lo}$, CD25⁻, CD2⁻), early large pre (LPre)-B (Lineage⁻, CD19⁺, B220⁺, sIgM⁻, CD43⁺, FSC$^{hi}$, CD2⁻), late LPre-B (Lineage⁻, CD19⁺, B220⁺, sIgM⁻, CD43⁺, FSC$^{hi}$, CD2⁺), small Pre (sPre)-B (Lineage⁻, CD19⁺, B220⁺, sIgM⁻, CD43$^{lo}$, FSC$^{lo}$, CD25⁺, CD2⁺), immature (Imm)-B (Lineage⁻, CD19⁺, B220$^{lo}$, sIgM⁺), and recirculating (Rec)-B (Lineage⁻, CD19⁺, B220$^{hi}$, sIgM⁺) based on the gating scheme shown in Supplementary Fig. 1a and a previous study[4], and isolated using FACSAriaII, or FACSAriaIII.

## RNA isolation and RT-qPCR

Cells were lysed in TRIzol Reagent, and the RNA was isolated according to the manufacturer's instructions. For the isolation of the RNA from sorted cells, RNA was isolated using RNA Clean & Concentrator-5 (Zymo Research). RNA was reverse transcribed by using ReverTra Ace (TOYOBO). cDNA was amplified by using PowerUp SYBR Green Master Mix (Applied Biosystems) and measured with StepOnePlus Real-Time PCR System (Applied Biosystems). To analyze mRNA expression, each RNA level was normalized with 18 s rRNA. The primers are listed in Supplementary Table 3.

## RNA-seq

Early LPre-B cells (*Upf1*-cKO or Ctrl) were sorted and lysed with TRIzol Reagent. The total RNA was isolated with RNA Clean & Concentrator-5 (Zymo Research). Ribosomal RNA was depleted from total RNA samples using NEBNext rRNA Depletion Kit v2. cDNA library was prepared using NEBNext Ultra II Directional RNA Library Prep Kit for Illumina (NEB) and sequenced on NextSeq 500 System (Illumina) according to the manufacturer's instructions. The acquired data was analyzed using Galaxy 2 (Afgan et al., 2018). Briefly, identified reads were mapped on the murine genome (mm10) using HISAT2 (single end, unstranded), and the mapped reads were counted using featureCounts.

For RNA-seq of *Igh*$^{B1-8hi}$, early LPre- and sPre-B cells were sorted and lysed with TRIzol Reagent. The total RNA was isolated with RNA Clean & Concentrator-5 (Zymo Research). Ribosomal RNA was depleted from total RNA samples using MGIEasy rRNA Depletion Kit V1.2 (MGI Tech). cDNA library was prepared using MGIEasy RNA Directional Library Prep Set V2.1 (MGI Tech) and sequenced on DNBSEQ-T7 (MGI Tech). The acquired reads were trimmed using Trim Galore (https://github.com/FelixKrueger/TrimGalore). Trimmed reads were aligned to the mouse genome (mm10) using HISAT2 (paired-end, unstranded), and the mapped reads were counted using featureCounts. The statistical analyses were performed using the edgeR package.

## RIP-seq of splenic B cells

Splenic B cells were isolated by depletion of CD43⁺ cells with autoMACS Pro Separator (Miltenyi Biotec) and stimulated with 5 μg/mL of anti-CD40 antibody (102810, BioLegend), 10 μg/mL of anti-IgM antibody (115-006-020, Jackson ImmunoResearch Inc), and 20 ng/mL of IL-4 (574302, BioLegend) for 72 h to activate the splenic B cells. After the stimulation, the B cells were lysed in RNA IP lysis buffer (0.5 % (vol/vol) Triton-X-100, 150 mM NaCl, 20 mM Tris-HCl (pH 8.0), cOmplete Mini Protease Inhibitor Cocktail without EDTA (Roche), PhosSTOP (Sigma) and 0.2 U/mL RNaseOut (Invitrogen)). A 5% volume of the cell lysate was collected as input and suspended with TRIzol reagent (Invitrogen). The rest lysate was incubated with 5 μg of anti-phospho-UPF1(07-1016, Merck) or normal rabbit IgG (MLB) bound to protein G magnetic beads (Invitrogen) for 3 h at 4 °C with lysates and beads were washed three times with wash buffer (0.1 % (vol/vol) Nonidet P-40, 150 mM NaCl, and 20 mM Tris-HCl (pH 8.0)). The precipitated RNAs were eluted from the beads using a Trizol reagent according to the manufacturer's instructions. RNA samples were purified with RNA Clean & Concentrator-5. cDNA library was prepared using NEBNext Ultra II Directional RNA Library Prep Kit for Illumina (NEB) and sequenced on NextSeq 500 System (Illumina) according to the manufacturer's instructions. Proteins were eluted from the beads using an SDS sample buffer and analyzed by immunoblot with the indicated antibodies.

In this study, we defined transcripts precipitated with anti-phospho-UPF1 (Log$_2$FC (anti-phospho-UPF1 / normal rabbit IgG) >= 1, adjusted $P$ value < 0.05, and Log$_2$FC (anti-phospho-UPF1 / input) >= 2, adjusted $P$ value < 0.05) as p-UPF1-binding target mRNAs (Fig. 7b, d, e, and f).

## Gene set enrichment analysis

Gene Set Enrichment Analysis was performed using GSEA software (version 4.3.2)[38,39]. Genes were ranked based on signal-to-noise ratio (*Upf1*-cKO vs Ctrl early LPre) or Log2FC (other comparisons) and the enrichment score was calculated ($p = 1$ (weighted)). First, we used hallmark gene sets in the Mouse Molecular Signatures Database (MSigDB)[40]. Next, we used all the gene sets and the identified gene sets were visualized using Cytoscape[41,42].

To perform enrichment analysis of *Zfp36l1/l2*-DCKO or *Cnot3*-cKO B cells, we analyzed the previously deposited transcriptome data[7,10]. These transcriptome data were respectively obtained from B cell progenitor during $V_H$-$D_H J_H$ recombination. The raw data was processed using Galaxy 2 as described above, and the genes were ranked based on Log2FC (KO vs Ctrl). Using the highly ($Log_2FC \geq 2$, adjP < 0.05) or poorly ($Log_2FC \leq -2$, adjP < 0.05) expressed genes in *Upf1*-cKO early LPre B cells the enrichment score was calculated ($p = 1$).

To examine the enrichment of p-UPF1-binding target RNAs in *Upf1*-cKO, *Cnot3*-cKO, or *Zfp36l1/l2* DCKO progenitor B cells, the genes identified in the RNA-seq were ranked based on Log2FC (KO vs Ctrl) and the enrichment score was calculated ($p = 1$). The p-UPF1-binding target mRNAs identified in splenic B cells (described above) were used as a gene set. To exclude the possibility that the genes expressed in splenic B cells were enriched in progenitor B cells of each genotype, the enrichment of whole genes identified in splenic B cells was also examined.

## Gene ontology enrichment analysis of UPF1-taget RNAs

Gene ontology of the identified UPF1-target RNAs was analyzed using metascape[43].

## V(D)J recombination analysis

Sorted early LPre-B cells were lysed with TRIzol reagent and the genomic DNA was isolated according to manufacturer's instructions. PCR analyses were performed using published primers as described previously[44,45] and DreamTaq DNA Polymerase (Thermo Fisher Scientific). Genome DNA samples isolated from wild-type mouse tail was used as negative control. The PCR products were electrophoresed on agarose gels and visualized with EtBr staining. The primers are listed in Supplementary Table 1.

## BrdU assay

The BrdU assay was performed with the FITC BrdU Flow Kit (BD Pharmingen) according to the manufacturer's instructions. Briefly, BrdU was intraperitoneally injected 24 hours before sacrificing the BM. The Obtained cells were permeabilized, fixed, stained with FITC-conjugated anti-BrdU antibody, and analyzed by Flow cytometry.

## Statistics & reproducibility

The sample size (n) for each group can be determined by the number of individual data points shown in each graph, which are all independent biological replicates. A Two-sided Student's *t* test was used to examine the statistical analysis for cell count, RT-qPCR. The statistical analyses for RNA-seq experiments were performed using limma or edgeR. No data were excluded from the analyses. No statistical methods were used to predetermine the sample size. The sample size was determined based on feasibility, pilot experiments, and according to published literature. Mouse and in vitro experiments were conducted and successfully replicated with sufficient numbers of mice or biological replicates as indicated in the Figure Legends section. All experiments were successfully repeated at least two times. Randomization was not required in this study because there is no statistic that requires randomization of samples.

## Reporting summary

Further information on research design is available in the Nature Portfolio Reporting Summary linked to this article.

## Data availability

The raw RNA sequence data generated in this study have been deposited in GEO under accession code GSE234830 (eLPre RNA-seq and RIP-seq) [https://www.ncbi.nlm.nih.gov/geo/query/acc.cgi?acc=GSE234830] and GSE264655 (B1-8^hi RNA-seq) [https://www.ncbi.nlm.nih.gov/geo/query/acc.cgi?acc=GSE264655]. The processed data reported in this paper are provided in the Supplementary Data files. All data supporting the findings of this study are present in the article and/or its Supplementary Information files. Source data are provided with this paper.

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

## Acknowledgements

We thank Y. Okumoto for secretarial assistance and all members of our laboratory for helpful discussion; T. Kondo and Y. Sando (Kyoto University) for technical assistance and RNA sequencing; H. Miyachi and S. Kitano (Kyoto University) for generating *Upf1*-flox mice; T. Kurosaki (Osaka University), M. Reth (University of Freiburg) and M. C. Nussenzweig (The Rockefeller University) for mice. This work was supported by grants from the Japan Society for the Promotion of Science (JSPS) KAKENHI (19H03488, 24K02224 to T.M. and 18H05278, 23H00402 to O.T.); Japan Science and Technology Agency (JST) FOREST Program (JPMJFR226E to T.M.) and Moonshot R&D (JPMJMS2025 to O.T.); Japan Agency for Medical Research and Development (AMED) (JP20gm4010002, JP21ae0121030, and JP20fk0108454 to O.T.); JSPS through the Core-to-Core Program (JPJSCCA20240006); grant-in-aid for Scientific Research on Innovative Areas 'Genome Science' (221S0002, 16H06279 to T.M.). T.M. was funded by the Takeda Science Foundation, Uehara Memorial Foundation, Shimizu Foundation for Immunology and Neuroscience, Naito Foundation, Senri Life Science Foundation, Nakajima Foundation, Mochida Memorial Foundation for Medical and Pharmaceutical Research, Fujiwara Memorial Foundation.

## Author contributions

N.I., T.M., and O.T. conceived the project and designed the experiments. N.I. performed most of the experiments. K.A., H.F., and M.Y. analyzed the RNA-seq data. L.W. helped with experiments. N.I., K.A., T.M., and O.T. wrote the manuscript. T.M. and O.T. supervised the project.

## Competing interests

The authors declare no competing interests.
