## [Peer Review File · Nature Communications]

UPF1 plays critical roles in early B cell developmentREVIEWER COMMENTS

Reviewer #1 (Remarks to the Author):

The paper by Iwai et al describes an analysis of the role of the RNA-Helicase UPF1. While the mRNA level of this gene is rather constant during B-cell development, protein expression is dramatically increased at the pre-B cell stage. Conditional KO of the UPF1 gene in B-lineage cells reveals a demand for this protein in the early to late pre-B cell stage. Gene expression analysis identified changes in the expression of genes associated with Interferon responses as well as IgH chain components. RIP-seq using splenic B-cells identified a set of genes with higher expression levels in the absence of UPF1 indicating that this protein is directly involved in the regulation of RNA levels in B-cell development. While deletion of *Ifnar1* could not rescue B-cell development in the absence of UPF1, expression of a functionally rearranged IgH chain rescued the early-pre-B to late pre-b transition. However, this was not sufficient to rescue subsequent developmental stages in the UPF1 deficient B-lineage cells. The authors suggest that the absence of functional VDJ recombination in the absence of UPF1 could be a result of that early-pre-B cells lacking this protein contain a large number of cells in S-phase, a stage non-compatible with functional rearrangements of the IgH gene.

The paper is well written and logical. Statistics and experimental design is clearly described allowing the reader to fully validate the data presented. Furthermore, the data clearly show that UPF1 is of large importance for normal B-cell development and that the early-to late pre-B cell transition can be rescued by expression of a functional IgH chain. The limitation of the paper resides in that the rescue is limited to one transition in B-cell development and that it is not clarified how UPF1 impact IgH VDJ recombination. The change in cell cycle is significant but small and unlikely explains the strong developmental defect observed.

Hence, while the basic finding that UPF1 is important for normal B-cell development, I do not find that the paper arrives in a conclusion regarding the actual role of this protein in B-cell development.

Major comments:

The authors use splenic cells for the RIP-seq and there seems to be high levels of pUPF1 in these cells (Figure 3B). Are the protein levels increased in the peripheral as compared to BM recirculating cells (Figure 1B). This would be important information. If not, could the large difference in protein expression levels result in that additional RNAs are target in the pre-B cells?

It is concerning that the recombined heavy chain only rescues one stage of development, especially as the UPF1 protein is hardly detected in the later developmental stages. A better understanding of this could be achieved by gene expression analysis of later developmental stages and peripheral B-lineage cells from mice where the pre-B stage is rescued by expression of the IgH chain.

Minor comments:

The title is misleading as VDJ recombination clearly is only relevant at the pre-B cell stage while UPF1 appear to have additional roles in other stages of development. I believe this should be indicated in the title.

The authors use a somewhat unconventional set of markers to define different B-cell progenitor subsets. The combined literature supports the authors use of markers but it would be beneficial for a non-expert reader if the marker selection would be motivated or if a reference to previous publications using this approach could be indicated.

In figure 3, it would be interesting to see how B-cell development look in the *Ifnar1* deficient mice.

Reviewer #2 (Remarks to the Author):

Early B-cell development is regulated not only by well-studied transcriptional program but also ever-growing post-transcriptional machineries, such as splicing regulator, RNA binding protein, RNA decay, RNA exosome degradation, etc. In this work, Noriki Iwai et al reported a new factor that governs early B cell development, particularly in VH-DHJH recombination. They provided clear genetic evidence showing B cell-specific Upf1 deletion in mice severely impeded early to late LPre-B cell transition. Transcriptomic analysis of early large B cells surprisingly revealed that strong type I IFN pathway was enriched in Upf1 KO cells, which seems unique compared to other reported model studying post-transcriptional regulation of early B cell development. The authors further nicely ruled out the involvement of IFN signaling in B cell deficiency by genetic crossing to *Ifnar1* KO mice. Introduction of genetically pre-arranged *Igh* could rescue the early B cell development to later Pre-B cells, but failed to develop further, suggesting Upf1 plays crucial roles in multiple stages through various mechanism. Overall, the data presented in the manuscript support the main conclusions, and the organization and presentation are commendable. Addressing the comments below will further strengthen the manuscript and contribute to a more comprehensive understanding of the role of UPF1 in early B cell development.

1. The authors observed an enhanced type I IFN pathway in Upf1 KO early B cells, and this could potentially be attributed to the activation of innate immune pathways, such as RNA-sensing, due to aberrant RNA decay and accumulation of immunogenic self-RNA. Additionally, only a limited number of ISGs were identified in late UPF1 RIP-seq, which raises questions. The authors should evaluate this possibility.
2. While the use of splenic B cells for UPF1 RIP-seq is understandable due to limitations in isolating early LPre-B cells, the dynamic change in UPF1 and p-UPF1 levels (Fig. 1b) suggests potential stage-specific UPF1 targets. It is crucial to acknowledge this limitation in the text and discuss the potential impact on the interpretation of the results.
3. The manuscript should provide a more detailed mechanistic explanation of how dysregulated UPF1 target genes are connected to the impairment of *Igh* VH-DHJH recombination.
4. The authors introduced genetically pre-arranged *Igh*, rescuing early B cell development to later LPre-B cells. To gain insights into the different roles of UPF1 in the two stages of B cell development, it is suggested that the authors perform transcriptomic analysis on B cells blocked at later LPre-B stage and compare it with the transcriptome of early LPre-B cells. This comparative analysis may provide valuable information about the specific contributions of UPF1 at distinct stages of B cell development and enhance the understanding of its regulatory mechanisms.

We thank the reviewers for their constructive criticisms. Below, please find our point-by-point responses to all comments and questions. We believe that we have examined each of the points raised and could respond thoroughly. Again, we would like to thank all the reviewers for being interested in our manuscript and for their thoughtful suggestions, which have definitely made this a stronger paper.

Reviewer #1:

The paper is well written and logical. Statistics and experimental design is clearly described allowing the reader to fully validate the data presented. Furthermore, the data clearly show that UPF1 is of large importance for normal B-cell development and that the early-to late pre-B cell transition can be rescued by expression of a functional IgH chain.

We thank the reviewer for finding this study well written and logical.

The limitation of the paper resides in that the rescue is limited to one transition in B-cell development and that it is not clarified how UPF1 impact IgH VDJ recombination. The change in cell cycle is significant but small and unlikely explains the strong developmental defect observed.

Hence, while the basic finding that UPF1 is important for normal B-cell development, I do not find that the paper arrives in a conclusion regarding the actual role of this protein in B-cell development.

We thank the reviewer for raising the constructive criticisms. The reviewer suggests us to provide mechanistic explanation of how dysregulated UPF1 target genes are connected to the impairment of *Igh* V_H-D_HJ_H recombination.

We demonstrated that *Upf1* deficiency resulted in an increase in B cells undergoing V_H-D_HJ_H recombination in the S-phase, potentially impeding V_H-D_HJ_H recombination. To further investigate the roles of UPF1 in controlling *Igh* recombination, we conducted a deeper investigation into the relationship between UPF1-regulated mRNAs and cell cycling by reanalyzing RNA-seq data from *Upf1*-cKO early LPre-B cells (Fig. 2). Our analysis revealed that a set of cell cycle-related genes exhibited heightened expression in *Upf1*-deficient early Lpre-B cells (new Fig. 5D). Additionally integrating the RIP-seq data for UPF1 (new Fig. 7), we identified *Nsl1*, *Dync1li2*, *Cenpo*, and *Rad9a*, as direct target mRNAs of UPF1, showing elevated expression in *Upf1*-deficient early

LPre-B cells (new Fig. 7I). These findings strongly suggest that dysregulation of UPF1 target genes associated with the cell cycle contributes to the observed cell cycle abnormalities and the loss of cell quiescence in *Upf1*-cKO early LPre-B cells.

Comparing LPre-B cells from *Upf1*-cKO mice and *Zfp3611/12*-DCKO (double conditional knockout) mice, we observed similarities in defects related to *Igh* V_H-D_HJ_H recombination (Fig. 2G, new 2H, 2I, 4A, and 4F), gene expression alteration (Fig. 5A), and cell cycle abnormalities (Fig. 5B and C). It has been shown that ZFP36L1/L2 interacts with and suppresses a set of mRNAs associated with cell cycling, and the increased expression of these mRNAs in *Zfp3611/12*-DCKO late pre-B cells contribute to defects in *Igh* recombination and B cell development by disrupting cell cycle quiescence (Galloway *et al.*, Science 352, 453-459, 2016). Considering that UPF1 was also revealed to directly target cell cycle-related mRNAs for suppression, it is plausible that UPF1 governs V_H-D_HJ_H recombination in a manner similar to ZFP36L1/L2.

Major comments:

The authors use splenic cells for the RIP-seq and there seems to be high levels of pUPF1 in these cells (Figure 3B). Are the protein levels increased in the peripheral as compared to BM recirculating cells (Figure 1B). This would be important information. If not, could the large difference in protein expression levels result in that additional RNAs are target in the pre-B cells?

We thank the reviewer for the valuable suggestion. As described in the methods section and Figure 7A, we utilized splenic B cells activated by treatment with IL-4, anti-IgM antibody and anti-CD40 antibody for the phosphorylated UPF1 (p-UPF1) RIP-seq analysis. In response to the reviewer's comment, we evaluated the abundance of phosphorylated UPF1 in early LPre-B (eLPre) and Recirculating-B (Rec-B) cells in the BM, as well as in splenic B cells stimulated with or without IL-4/IgM/CD40, using immunoblot analysis.

Consistent with the data presented in Figure 1B, the levels of p-UPF1 were lower in Rec-B cells compared to eLPre cells in the BM (Figure R1, below). Although the level of p-UPF1 was initially low in splenic naïve B cells, it increased following stimulation with IL-4/IgM/CD40 (Figure R1). While the abundance of p-UPF1 in stimulated splenic B cells was not as high as that of early LPre-B cells (Figure R1), we posit that the UPF1 phosphorylation status is adequate for conducting RIP-seq analysis using anti-p-UPF1 Ab. Indeed, the RIP-seq analysis identified well-known UPF1 targeted genes such as *Dtit3*,

Smg5, *Atf3* and *Gadd45b* (Figure 7B).

However, considering the differences in phosphorylated UPF1 levels and the transcriptome between BM eLPre cells and stimulated splenic B cells, it is likely that there are additional UPF1-binding mRNAs specific to early LPre-B cells. Nevertheless, given the remarkable correlation between UPF1-binding mRNAs in splenic B cells and genes upregulated in BM eLPre cells under *Upf1* deficiency, it is evident that a proportion of mRNAs are commonly regulated by UPF1 throughout B cell differentiation and activation. We discussed the presence of potential UPF1-binding mRNAs specific to early LPre-B cells in the discussion section (page 30).

Notably, we presented the RIP-seq data in new Figure 7. This decision was made to facilitate the discussion of UPF1-target mRNAs in sPre-B cells, allowing for a comparison of the RIP-seq data with the transcriptome analysis using sPre-B cells lacking UPF1 in the presence of the pre-rearranged *Igh* allele (new Figure 6), which was conducted in response to the reviewers' suggestion.

*Used for RIP-Seq (in Figure 7)

Figure R1. The expression levels of phosphorylated UPF1 in BM and splenic B cells
The expression levels of phosphorylated UPF1 in indicated B cells were analyzed by the immunoblot analysis.

It is concerning that the recombined heavy chain only rescues one stage of development, especially as the UPF1 protein is hardly detected in the later developmental stages. A better understanding of this could be achieved by gene expression analysis of later developmental stages and peripheral B-lineage cells from mice where the pre-B stage is rescued by expression of the IgH chain.

We thank the reviewer for the thoughtful suggestion. In response to the reviewer's suggestion, we evaluated the transcriptome of sPre-B cells lacking UPF1 in the presence

of the pre-rearranged *Igh* allele (*Upfl*-cKO/*Igh*^{B1-8hi}) and control (Ctrl/*Igh*^{B1-8hi}) sPre-B cells (new Supplementary Table 2). By comparing transcriptomic profiles between early LPre- and sPre-B cells from Ctrl/*Igh*^{B1-8hi} (control) mice, we noted dynamic changes in gene expression during transition from early LPre- to sPre-B cells: the downregulation of 1,533 genes (Log2FC ≤ -2, adjP < 0.05) and upregulation of 847 genes (Log2FC ≥ 2, adjP < 0.05) (new Fig. 6C). Interestingly, the genes that showed decreased expression levels during the differentiation from early LPre- to sPre-B cells in control cells exhibited higher expression in *Upfl*-cKO/*Igh*^{B1-8hi} sPre-B cells than Ctrl/*Igh*^{B1-8hi} sPre-B cells (new Fig. 6D).

Further GSEA revealed the enrichment of gene sets associated with cell cycle progression, such as E2F_TARGETS (adj. P = 0.087), as well as UNFOLDED_PROTEIN_RESPONSE (UPR) (adj. P = 0.37, P = 0.021) in *Upfl*-cKO/*Igh*^{B1-8hi} sPre-B cells compared to Ctrl/*Igh*^{B1-8hi} cells (new Fig. 6E). In control mice, during the differentiation from early LPre- to sPre-B cells, a subset of gene sets related to cell cycling-including E2F target, G2M_Checkpoint, and MYC target-exhibited downregulation (new Supplementary Figure 6A, B). These alterations align with the transition from actively cycling large Pre-B cells to quiescent small Pre-B cells. The enrichment of the E2F target gene set in sPre-B cells, but not early LPre-B cells, lacking UPF1 (cKO/*Igh*^{B1-8hi}) (new Fig. 6E) suggests a potential requirement of UPF1 in driving the downregulation of cell cycle-related gene expression during the transition from early LPre- to sPre-B stage.

In contrast, consistent with results of PCR assay for the *Ig light chain*, the expression levels of v-region transcripts from *Ig light chains* (*Igkv* and *Iglv*) were largely similar between *Upfl*-cKO/*Igh*^{B1-8hi} and Ctrl/*Igh*^{B1-8hi} sPre-B cells (new Fig. 6F). The finding suggests that UPF1 is dispensable for the recombination at *Ig light chain* locus.

Notably, the gene sets “INTERFERON ALPHA RESPONSE” and “INTERFERON GAMMA RESPONSE”, which were significantly enriched in cKO early LPre-B cells, did not show enrichment in early LPre-B cells in the absence of UPF1 with the pre-rearranged *Igh* allele (cKO/*Igh*^{B1-8hi}) (new Fig. 6G). Consistently, genes related to the IFN response were only modestly elevated in cKO/*Igh*^{B1-8hi} early LPre-B cells, whereas these genes were more robustly increased in cKO/*Igh*^{WT} early LPre-B cells (new Fig. 6H). These data suggest that the *Igh* pre-rearrangement prevented the aberrant expression of IFN response-related genes in *Upfl*-deficient early LPre-B cells. In contrast, IFN response-related genes were upregulated in sPre-B cells from cKO/*Igh*^{B1-8hi} compared with Ctrl/*Igh*^{B1-8hi} mice (new Fig. 6H). For instance, ISGs, such as *Ifi44* and *Ddx60*, started to elevate in cKO/*Igh*^{B1-8hi} sPre-B cells (new Supplementary Figure 6C),

suggesting that the *Igh* pre-rearrangement failed to suppress the expression of IFN response-related genes in sPre-B cells lacking UPF1. These results imply that UPF1 plays a role on preventing the abnormal expression of IFN response-related genes not only in early LPre-B stage but also in the sPre-B stage.

The UPR acts as critical a checkpoint in B cell development by regulating the maintenance of BCR expression and proper transport to the cell surface^{29, 30, 31}. Genes associated with the UPR were highly enriched in both early LPre-B (adj. P = 0.0075, P < 0.0001) and sPre-B cells (adj. P = 0.37, P = 0.021) from cKO/*Igh*^{B1-8hi} mice (new Fig. 6E, G). We also found a comparable number of UPR genes affected by UPF1 depletion in *Igh*^{WT} early LPre, *Igh*^{B1-8hi} early LPre-, and *Igh*^{B1-8hi} sPre-B cells (new Fig. 6I). These data suggest that, unlike IFN-related genes, the omission of *Igh* recombination through B1-8^{hi} knock-in did not mitigate the abnormally high expression of UPR genes in *Upfl*-deficient B cells. Some of the UPR genes were downregulated during the differentiation from early LPre - to sPre-B stage in *Upfl*-Ctrl/*Igh*^{B1-8hi} B cells (GSEA: adj.P = 0.060) (new Fig. 6J and Supplementary Figure 6A), suggesting a delayed downregulation of UPR genes during the transition from early LPre- to sPre-B stage in *Upfl*-deficient B cells.

We think that these data demonstrate that UPF1 is critical for ensuring the transcriptome shift from the early LPre- to sPre-B stage by preventing abnormal expression of genes related to cell cycle, IFN response, and the UPR. We described these new findings in the new Figure 6. Considering the significance of UPF1-mediated regulation of cell cycle-related genes in early LPre B cells, we opted to present the GSEA results for E2F TARGETS, initially featured in Supplementry Figure 2, within the main Figure 2E and F in the revised manuscript.

We also discussed the potential contribution of elevated UPR gene expression in *Upfl*-deficient sPre-B cells on the regulation of BCR expression and its proper transport to the cell surface, which could potentially hinder the differentiation process from sPre-B cells to immature B cells (page 30).

Minor comments:

The title is misleading as VDJ recombination clearly is only relevant at the pre-B cell stage while UPF1 appear to have additional roles in other stages of development. I believe this should be indicated in the title.

We appreciate the suggestion from the reviewer. As noted by the reviewer, the roles of UPF1 in the control of B cell development extend beyond the regulation of the VDJ

recombination. Therefore, according to the reviewer's suggestion, we have revised the title of this paper to "Critical roles of UPF1 in early B cell development".

The authors use a somewhat unconventional set of markers to define different B-cell progenitor subsets. The combined literature supports the authors use of markers but it would be beneficial for a non-expert reader if the marker selection would be motivated or if a reference to previous publications using this approach could be indicated.

We thank the reviewer for the valuable suggestion. To characterize B cell progenitor subsets, we carefully selected markers based on a previous study that elucidated the role of METTL14 in B cell development (Zheng, Z. et al., *Cell Rep.* 31, 107819, 2020). In addition, we employed CD2, a surface antigen induced by the presence of cytoplasmic mu-chain (Sen, J. et al. *J. Immunol.* 144, 2925-2930, 1990), to differentiate between early and late LPre-B cells. We are confident that the combination of these markers allowed us to delineate Pre-B cell populations both before and after successful *Igh* recombination. In response to the reviewer's suggestion, we have provided a detailed description of our selection of markers and gating strategy, and included citations to these papers in the methods section.

*In figure 3, it would be interesting to see how B-cell development look in the *Ifnar1* deficient mice.*

We thank the reviewer for the insightful suggestion. Following the reviewer's suggestion, we investigated the potential impact of the absence of type I IFN receptor (IFNAR1) on B cell development. We conducted flow cytometry analysis on BM cells from control (*Upf1^{Flox/Flox} Mbl^{+/+} Ifnar1^{+/-}*) and *Ifnar1*-KO (*Upf1^{Flox/Flox} Mbl^{+/+} Ifnar1^{-/-}*) mice. Notably, these mice express UPF1 due to the absence of Cre expression. The analysis did not reveal any discernible difference in B cell development within the BM between control and *Ifnar1*-KO mice (new Supplementary Figure 3). Additionally, a previous study also indicated the absence of significant differences between control and *Ifnar1*-deficient mice across various stages of B cell differentiation (Domeier, P.P. et al., *Cell Rep.* 24, 406-418, 2018). This study included populations both before and after *Igh* recombination, with normal observations of splenic B cells illustrated in Figure S1 of the aforementioned paper. Furthermore, Spurrier, M.A. et al. presented data indicating

normal splenic B cell development in *Ifnar1*^{-/-} mice through flow cytometry analysis (Figure 2 of Spurrier, M.A. *et al.*, *J. Immunol.* 210, 148-157, 2023). Our finding, combined with these prior studies, collectively support the conclusion that the loss of type I IFN signaling does not impact B cell development in the BM and spleen.

Reviewer #2:

Overall, the data presented in the manuscript support the main conclusions, and the organization and presentation are commendable. Addressing the comments below will further strengthen the manuscript and contribute to a more comprehensive understanding of the role of UPF1 in early B cell development.

We thank the reviewer for finding this study well organized and commendable.

*1. The authors observed an enhanced type I IFN pathway in *Upf1* KO early B cells, and this could potentially be attributed to the activation of innate immune pathways, such as RNA-sensing, due to aberrant RNA decay and accumulation of immunogenic self-RNA. Additionally, only a limited number of ISGs were identified in late UPF1 RIP-seq, which raises questions. The authors should evaluate this possibility.*

We thank the reviewer for the thoughtful suggestion. As the reviewer pointed out, UPF1 deficiency potentially induces the activation of innate immune pathways due to the accumulation of immunogenic self-RNAs resulting from impaired RNA decay pathways. It is well established that such self-RNAs can be recognized by cytosolic double-stranded RNA sensors MDA5 and RIG-I, which utilize MAVS as the adaptor to trigger downstream signaling pathways (Liddicoat BJ, *et al.* *Science.* 349, 1115-20, 2015, Crow YJ, *et al.* *Nat Rev Immunol.* 7, 429-40, 2015, Uehata T and Takeuchi O. *Cells* 9, 1701, 2020). Therefore, we evaluated the role of RIG-I-like receptors in B cell development under UPF1 deficiency by generating *Mavs* (mitochondrial antiviral signaling)-KO mice crossed with *Upf1*-cKO mice. We observed that the spleen size and splenic B cell populations were not rescued in *Upf1/Mavs*-DKO mice (*Upf1*^{Flox/Flox} *Mb1*^{Cre/+} *Mavs*^{-/-}) compared to *Upf1*-cKO mice (Fig. R2 A, B below). In addition, the defects in B cell differentiation from early to late LPre-B cells transition in *Upf1/Mavs*-DKO mice remained similar to those observed in *Upf1*-cKO mice and *Upf1/Ifnar1*-DKO mice (Fig. R2 C, D). Furthermore, the expression of ISGs such as *Mx1* and *Mx2* was not abrogated

in *Upf1/Mavs*-DKO early LPre-B cells (Fig. R2 E). These results suggest that the RIG-I-like receptor signaling is not critical for the activation of IFN responses or the perturbation of early to late LPre-B cell development in *Upf1*-deficient B cells.

Therefore, it is possible that receptors other than RIG-I-like receptors might trigger the activation of IFN responses in *Upf1*-cKO mice. Alternatively, direct regulation of ISGs by UPF1 could contribute to the transcriptomic changes observed in early LPre-B cells under *Upf1* deficiency. Further studies are needed to clarify UPF1's mechanism of controlling ISGs in B cell progenitors. However, it is worth noting that the significance of this control seems limited, considering that even the lack of type I IFN signaling did not rescue the B cell differentiation defects in *Upf1* deficiency.

Figure R2 The absence of *Mavs* does not alleviate the deficiencies in B cell development observed in *Upf1*-cKO mice

- (A) Spleens derived from indicated mice. Results are representative of two independent experiments.
- (B) Flow cytometry plots of indicated populations in the splenocytes from specified mice. Results are representative of two independent experiments.
- (C) Flow cytometry plots showing indicated populations in the BM from specified mice. Results are representative of two independent experiments.
- (D) Flow cytometry plots of BM FVD⁻CD19⁺B220⁺sIgM⁻ cells from indicated mice. Results are representative of two independent experiments.
- (E) The expression of Gas5, Mx1 and Mx2 mRNAs in early LPre-B cells from indicated mice.

2. While the use of splenic B cells for UPF1 RIP-seq is understandable due to limitations in isolating early LPre-B cells, the dynamic change in UPF1 and p-UPF1 levels (Fig. 1b) suggests potential stage-specific UPF1 targets. It is crucial to acknowledge this limitation in the text and discuss the potential impact on the interpretation of the results.

As pointed out by the reviewer, the efficiency of identifying UPF1-targeted mRNAs in RIP-seq analysis could be compromised if the expression level of phosphorylated UPF1 is significantly lower in splenic activated B cells used for the analysis compared to early LPre-B cells. However, our response to a similar comment raised by Reviewer #1 demonstrated that the levels of phosphorylated UPF1 in splenic activated B cells were notably higher than in Recirculating-B (Rec-B) cells in the BM (Figure R1 above). Further, our RIP-seq analysis successfully identified UPF1 target genes associated with immune responses, cell cycle regulation, and IFN response-related genes in early LPre-B cells (new Fig. 7 and new Supplementary Fig. 7D), indicating the efficacy of our RIP-seq approach. Nevertheless, it remains unclear whether UPF1 targets 40 to 60% of highly expressed genes in *Upf1*-deficient progenitor B cells, as the RNAs identified in RIP-seq using splenic B cells did not cover all the genes expressed in early LPre- or sPre-B cells. As suggested by the reviewer, there is a possibility of additional target mRNAs specifically expressed in early LPre- and sPre-B cells that are regulated by UPF1. We have discussed this possibility in the Discussion section as recommended.

3. The manuscript should provide a more detailed mechanistic explanation of how dysregulated UPF1 target genes are connected to the impairment of Igh VH-DHJH

recombination.

We thank the reviewer for this important suggestion. Our study demonstrates that *Upf1*-deficiency resulted in an increase in B cells undergoing V_H-D_HJ_H recombination in the S-phase, potentially impeding V_H-D_HJ_H recombination. In response to reviewer's concern, we conducted a deeper investigation into the relationship between UPF1-regulated mRNAs and cell cycling by reanalyzing RNA-seq data from *Upf1*-cKO early LPre-B cells (Fig. 2). Our analysis revealed that a set of cell cycle-related genes exhibited heightened expression in *Upf1*-deficient early Lpre-B cells (new Fig. 5D). Additionally integrating the RIP-seq data for UPF1 (Fig. 7), we identified *Ns11*, *Dync1li2*, *Cenpo*, and *Rad9a*, as direct target mRNAs of UPF1, showing elevated expression in *Upf1*-deficient early LPre-B cells (new Fig. 7I). These findings strongly suggest that dysregulation of UPF1 target genes associated with the cell cycle contributes to the observed cell cycle abnormalities and the loss of cell quiescence in *Upf1*-cKO early LPre-B cells.

Comparing Pre-B cells from *Upf1*-cKO mice and *Zfp3611/12*-DCKO (double conditional knockout) mice, we observed similarities in defects related to *Igh* V_H-D_HJ_H recombination (Fig. 2G, new 2H, 2I, 4A and 4F), gene expression alteration (Fig. 5A), and cell cycle abnormalities (Fig. 5B and C). It has been shown that ZFP36L1/L2 interacts with and suppresses a set of mRNAs associated with cell cycling, and the increased expression of these mRNAs in *Zfp3611/12*-DCKO late pre-B cells contribute to defects in *Igh* recombination and B cell development by disrupting cell cycle quiescence (Galloway *et al.*, Science 352, 453-459, 2016). Considering that UPF1 was also revealed to directly target cell cycle-related mRNAs for suppression, it is plausible that UPF1 governs V_H-D_HJ_H recombination in a manner similar to ZFP36L1/L2. We showed the results of the new analysis in the results section.

4. The authors introduced genetically pre-arranged Igh, rescuing early B cell development to later LPre-B cells. To gain insights into the different roles of UPF1 in the two stages of B cell development, it is suggested that the authors perform transcriptomic analysis on B cells blocked at later LPre-B stage and compare it with the transcriptome of early LPre-B cells. This comparative analysis may provide valuable information about the specific contributions of UPF1 at distinct stages of B cell development and enhance the understanding of its regulatory mechanisms.

We thank the reviewer for the thoughtful suggestion. According to the reviewer's

suggestion, we conducted RNA-seq analysis using early LPre- and sPre-B cells derived from cKO/*Igh*^{B1-8hi} and Ctrl/*Igh*^{B1-8hi} mice (new Supplementary Table 2). By comparing transcriptomic profiles between early LPre- and sPre-B cells from Ctrl/*Igh*^{B1-8hi} (control) mice, we noted dynamic changes in gene expression during transition from early LPre- to sPre-B cells: the downregulation of 1,533 genes (Log2FC ≤ -2, adjP < 0.05) and upregulation of 847 genes (Log2FC ≥ 2, adjP < 0.05) (new Fig. 6C). Interestingly, the genes that showed decreased expression levels during the differentiation from early LPre- to sPre-B cells in control cells exhibited higher expression in *Upf1*-cKO/*Igh*^{B1-8hi} sPre-B cells than Ctrl/*Igh*^{B1-8hi} sPre-B cells (new Fig. 6D).

Further GSEA revealed the enrichment of gene sets associated with cell cycle progression, such as E2F_TARGETS (adj. P = 0.087), as well as UNFOLDED_PROTEIN_RESPONSE (UPR) (adj. P = 0.37, P = 0.021) in *Upf1*-cKO/*Igh*^{B1-8hi} sPre-B cells compared to Ctrl/*Igh*^{B1-8hi} cells (new Fig. 6E). In control mice, during the differentiation from early LPre- to sPre-B cells, a subset of gene sets related to cell cycling-including E2F target, G2M_Checkpoint, and MYC target-exhibited downregulation (new Supplementary Figure 6A, B). These alterations align with the transition from actively cycling large Pre-B cells to quiescent small Pre-B cells. The enrichment of the E2F target gene set in sPre-B cells, but not early LPre-B cells, lacking UPF1 (cKO/*Igh*^{B1-8hi}) (new Fig. 6E) suggests a potential requirement of UPF1 in driving the downregulation of cell cycle-related gene expression during the transition from early LPre- to sPre-B stage.

In contrast, consistent with results of PCR assay for the *Ig light chain*, the expression levels of v-region transcripts from *Ig light chains* (*Igkv* and *Iglv*) were largely similar between *Upf1*-cKO/*Igh*^{B1-8hi} and Ctrl/*Igh*^{B1-8hi} sPre-B cells (new Fig. 6F). The finding suggests that UPF1 is dispensable for the recombination at *Ig light chain* locus.

Notably, the gene sets “INTERFERON ALPHA RESPONSE” and “INTERFERON GAMMA RESPONSE”, which were significantly enriched in cKO early LPre-B cells, did not show enrichment in early LPre-B cells in the absence of UPF1 with the pre-rearranged *Igh* allele (cKO/*Igh*^{B1-8hi}) (new Fig. 6G). Consistently, genes related to the IFN response were only modestly elevated in cKO/*Igh*^{B1-8hi} early LPre-B cells, whereas these genes were more robustly increased in cKO/*Igh*^{WT} early LPre-B cells (new Fig. 6H). These data suggest that the *Igh* pre-rearrangement prevented the aberrant expression of IFN response-related genes in *Upf1*-deficient early LPre-B cells. In contrast, IFN-related genes were upregulated in sPre-B cells from cKO/*Igh*^{B1-8hi} compared with Ctrl/*Igh*^{B1-8hi} mice (new Fig. 6H). For instance, ISGs, such as *Ifi44* and *Ddx60*, started to elevate in cKO/*Igh*^{B1-8hi} sPre-B cells (new Supplementary Figure 6C), suggesting that the

Igh pre-rearrangement failed to suppress the expression of IFN response-related genes in sPre-B cells lacking UPF1. These results imply that UPF1 plays a role on preventing the abnormal expression of IFN response-related genes not only in early LPre-B stage but also in the sPre-B stage.

The UPR acts as critical a checkpoint in B cell development by regulating the maintenance of BCR expression and proper transport to the cell surface^{29, 30, 31}. Genes associated with the UPR were highly enriched in both early LPre-B (adj. P = 0.0075, P < 0.0001) and sPre-B cells (adj. P = 0.37, P = 0.021) from cKO/*Igh*^{B1-8hi} mice (new Fig. 6E, G). We also found a comparable number of UPR genes affected by UPF1 depletion in *Igh*^{WT} early LPre, *Igh*^{B1-8hi} early LPre-, and *Igh*^{B1-8hi} sPre-B cells (new Fig. 6I). These data suggest that, unlike IFN-related genes, the omission of *Igh* recombination through B1-8^{hi} knock-in did not mitigate the abnormally high expression of UPR genes in *Upfl*-deficient B cells. Some of the UPR genes were downregulated during the differentiation from early LPre - to sPre-B stage in *Upfl*-Ctrl/*Igh*^{B1-8hi} B cells (GSEA: adj.P = 0.060) (new Fig. 6J and Supplementary Figure 6A), suggesting a delayed downregulation of UPR genes during the transition from early LPre- to sPre-B stage in *Upfl*-deficient B cells.

We think that these data demonstrate that UPF1 is critical for ensuring the transcriptome shift from the early LPre- to sPre-B stage by preventing abnormal expression of genes related to cell cycle, IFN response, and the UPR. We described these new findings in the new Figure 6.

Considering the significance of UPF1-mediated regulation of cell cycle-related genes in early LPre B cells, we opted to present the GSEA results for E2F TARGETS, initially featured in Supplementry Figure 2, within the main Figure 2E and F in the revised manuscript.

REVIEWERS' COMMENTS

Reviewer #1 (Remarks to the Author):

The authors have adressed my concerns. Thank you.

Reviewer #2 (Remarks to the Author):

The authors performed additional experiments to address my comments seriously. The revised manuscript was substantially improved. I have not further comments.

We would like to thank all the reviewers for carefully evaluating our manuscript. We are delighted that our previous responses and revisions satisfied reviewers.

Reviewer #1:

The authors have adressed my concerns. Thank you.

We thank the reviewer for acknowledging that we have addressed all the concerns raised.

Reviewer #2:

The authors performed additional experiments to address my comments seriously. The revised manuscript was substantially improved. I have not further comments.

We thank the reviewer for acknowledging that we have addressed all the concerns raised.

The Editor:

We thank the editor for suggestions. We revised the manuscript according to the Author Checklist.